# Zero-Shot Image Restoration Using Denoising Diffusion Null-Space Model

**Yinhuai Wang**[1*]**, Jiwen Yu**[1*]**, Jian Zhang**[1,2†]

[1]Peking University Shenzhen Graduate School, [2]Peng Cheng Laboratory
{yinhuai; yujiwen}@stu.pku.edu.cn, zhangjian.sz@pku.edu.cn

## ABSTRACT

Most existing Image Restoration (IR) models are task-specific, which can not be generalized to different degradation operators. In this work, we propose the Denoising Diffusion Null-Space Model (DDNM), a novel zero-shot framework for arbitrary linear IR problems, including but not limited to image super-resolution, colorization, inpainting, compressed sensing, and deblurring. DDNM only needs a pre-trained off-the-shelf diffusion model as the generative prior, without any extra training or network modifications. By refining only the null-space contents during the reverse diffusion process, we can yield diverse results satisfying both data consistency and realness. We further propose an enhanced and robust version, dubbed DDNM$^+$, to support noisy restoration and improve restoration quality for hard tasks. Our experiments on several IR tasks reveal that DDNM outperforms other state-of-the-art zero-shot IR methods. We also demonstrate that DDNM$^+$ can solve complex real-world applications, *e.g.*, old photo restoration.

## 1 INTRODUCTION

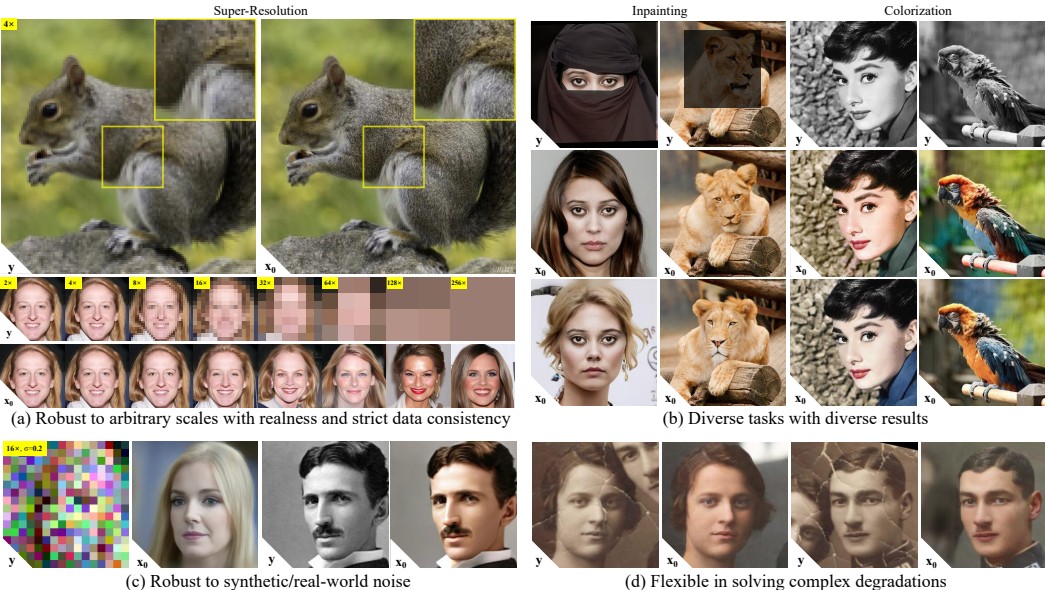

(a) Robust to arbitrary scales with realness and strict data consistency

(b) Diverse tasks with diverse results

(c) Robust to synthetic/real-world noise

(d) Flexible in solving complex degradations

Figure 1: **We use DDNM$^+$ to solve various image restoration tasks in a zero-shot way**. Here we show some of the results that best characterize our method, where $\mathbf{y}$ is the input degraded image and $\mathbf{x}_0$ represents the restoration result. Part (a) shows the results of DDNM$^+$ on image super-resolution (SR) from scale $2\times$ to extreme scale $256\times$. Note that DDNM$^+$ assures strict data consistency. Part (b) shows multiple results of DDNM$^+$ on inpainting and colorization. Part (c) shows the results of DDNM$^+$ on SR with synthetic noise and colorization with real-world noise. Part (d) shows the results of DDNM$^+$ on old photo restoration. All the results here are yielded in a **zero-shot** way.

*Equal contribution. † Corresponding author. Code is available at https://github.com/wyhuai/DDNM. This work was supported in part by Shenzhen Research Project under Grant JCYJ20220531093215035 and Grant JSGGZD20220822095800001.

Image Restoration (IR) is a long-standing problem due to its extensive application value and its ill-posed nature (Richardson, 1972; Andrews & Hunt, 1977). IR aims at yielding a high-quality image $\hat{\mathbf{x}}$ from a degraded observation $\mathbf{y} = \mathbf{A}\mathbf{x} + \mathbf{n}$, where $\mathbf{x}$ stands for the original image and $\mathbf{n}$ represents a non-linear noise. $\mathbf{A}$ is a known linear operator, which may be a bicubic downsampler in image super-resolution, a sampling matrix in compressed sensing, or even a composite type. Traditional IR methods are typically model-based, whose solution can be usually formulated as:

$$\hat{\mathbf{x}} = \arg\min_{\mathbf{x}} \frac{1}{2\sigma^2} ||\mathbf{A}\mathbf{x} - \mathbf{y}||_2^2 + \lambda\mathcal{R}(\mathbf{x}). \tag{1}$$

The first data-fidelity term $\frac{1}{2\sigma^2}||\mathbf{A}\mathbf{x} - \mathbf{y}||_2^2$ optimizes the result toward data consistency while the second image-prior term $\lambda\mathcal{R}(\mathbf{x})$ regularizes the result with formulaic prior knowledge on natural image distribution, e.g., sparsity and Tikhonov regularization. Though the hand-designed prior knowledge may prevent some artifacts, they often fail to bring realistic details.

The prevailing of deep neural networks (DNN) brings new patterns of solving IR tasks (Dong et al., 2015), which typically train an end-to-end DNN $\mathcal{D}_{\boldsymbol{\theta}}$ by optimizing network parameters $\boldsymbol{\theta}$ following

$$\arg\min_{\boldsymbol{\theta}} \sum_{i=1}^{N} ||\mathcal{D}_{\boldsymbol{\theta}}(\mathbf{y}_i) - \mathbf{x}_i||_2^2, \tag{2}$$

where $N$ pairs of degraded image $\mathbf{y}_i$ and ground truth image $\mathbf{x}_i$ are needed to learn the mapping from $\mathbf{y}$ to $\mathbf{x}$ directly. Although end-to-end learning-based IR methods avoid explicitly modeling the degradation $\mathbf{A}$ and the prior term in Eq. 1 and are fast during inference, they usually lack interpretation. Some efforts have been made in exploring interpretable DNN structures (Zhang & Ghanem, 2018; Zhang et al., 2020), however, they still yield poor performance when facing domain shift since Eq. 2 essentially encourage learning the mapping from $\mathbf{y}_i$ to $\mathbf{x}_i$. For the same reason, the end-to-end learning-based IR methods usually need to train a dedicated DNN for each specific task, lacking generalizability and flexibility in solving diverse IR tasks. The evolution of generative models (Goodfellow et al., 2014; Bahat & Michaeli, 2014; Van Den Oord et al., 2017; Karras et al., 2019; 2020; 2021) further pushes the end-to-end learning-based IR methods toward unprecedented performance in yielding realistic results (Yang et al., 2021; Wang et al., 2021; Chan et al., 2021; Wang et al., 2022). At the same time, some methods (Menon et al., 2020; Pan et al., 2021) start to leverage the latent space of pretrained generative models to solve IR problems in a zero-shot way. Typically, they optimize the following objective:

$$\arg\min_{\mathbf{w}} \frac{1}{2\sigma^2} ||\mathbf{A}\mathcal{G}(\mathbf{w}) - \mathbf{y}||_2^2 + \lambda\mathcal{R}(\mathbf{w}), \tag{3}$$

where $\mathcal{G}$ is the pretrained generative model, $\mathbf{w}$ is the latent code, $\mathcal{G}(\mathbf{w})$ is the corresponding generative result and $\mathcal{R}(\mathbf{w})$ constrains $\mathbf{w}$ to its original distribution space, e.g., a Gaussian distribution. However, this type of method often struggles to balance realness and data consistency.

The Range-Null space decomposition (Schwab et al., 2019; Wang et al., 2023) offers a new perspective on the relationship between realness and data consistency: the data consistency is only related to the range-space contents, which can be analytically calculated. Hence the data term can be strictly guaranteed, and the key problem is to find proper null-space contents that make the result satisfying realness. We notice that the emerging diffusion models (Ho et al., 2020; Dhariwal & Nichol, 2021) are ideal tools to yield ideal null-space contents because they support explicit control over the generation process.

In this paper, we propose a novel zero-shot solution for various IR tasks, which we call the Denoising Diffusion Null-Space Model (DDNM). By refining only the null-space contents during the reverse diffusion sampling, our solution only requires an off-the-shelf diffusion model to yield realistic and data-consistent results, without any extra training or optimization nor needing any modifications to network structures. Extensive experiments show that DDNM outperforms state-of-the-art zero-shot IR methods in diverse IR tasks, including super-resolution, colorization, compressed sensing, inpainting, and deblurring. We further propose an enhanced version, DDNM$^+$, which significantly elevates the generative quality and supports solving noisy IR tasks. Our methods are free from domain shifts in degradation modes and thus can flexibly solve complex IR tasks with real-world degradation, such as old photo restoration. Our approaches reveal a promising new path toward solving IR tasks in zero-shots, as the data consistency is analytically guaranteed, and the realness

is determined by the pretrained diffusion models used, which are rapidly evolving. Fig. 1 provides some typical applications that fully show the superiority and generality of the proposed methods.

**Contributions.** (1) **In theory**, we reveal that a pretrained diffusion model can be a zero-shot solver for linear IR problems by refining only the null-space during the reverse diffusion process. Correspondingly, we propose a unified theoretical framework for arbitrary linear IR problems. We further extend our method to support solving noisy IR tasks and propose a *time-travel* trick to improve the restoration quality significantly; (2) **In practice**, our solution is the first that can decently solve diverse linear IR tasks with arbitrary noise levels, in a zero-shot manner. Furthermore, our solution can handle composite degradation and is robust to noise types, whereby we can tackle challenging real-world applications. Our proposed DDNMs achieve state-of-the-art zero-shot IR results.

## 2 BACKGROUND

### 2.1 REVIEW THE DIFFUSION MODELS

We follow the diffusion model defined in denoising diffusion probabilistic models (DDPM) (Ho et al., 2020). DDPM defines a $T$-step forward process and a $T$-step reverse process. The forward process slowly adds random noise to data, while the reverse process constructs desired data samples from the noise. The forward process yields the present state $\mathbf{x}_t$ from the previous state $\mathbf{x}_{t-1}$:

$$q(\mathbf{x}_t|\mathbf{x}_{t-1}) = \mathcal{N}(\mathbf{x}_t; \sqrt{1-\beta_t}\mathbf{x}_{t-1}, \beta_t\mathbf{I}) \quad i.e., \quad \mathbf{x}_t = \sqrt{1-\beta_t}\mathbf{x}_{t-1} + \sqrt{\beta_t}\boldsymbol{\epsilon}, \quad \boldsymbol{\epsilon} \sim \mathcal{N}(0, \mathbf{I}), \tag{4}$$

where $\mathbf{x}_t$ is the noised image at time-step $t$, $\beta_t$ is the predefined scale factor, and $\mathcal{N}$ represents the Gaussian distribution. Using reparameterization trick, it becomes

$$q(\mathbf{x}_t|\mathbf{x}_0) = \mathcal{N}(\mathbf{x}_t; \sqrt{\bar{\alpha}_t}\mathbf{x}_0, (1-\bar{\alpha}_t)\mathbf{I}) \quad \text{with} \quad \alpha_t = 1-\beta_t, \quad \bar{\alpha}_t = \prod_{i=0}^{t}\alpha_i. \tag{5}$$

The reverse process aims at yielding the previous state $\mathbf{x}_{t-1}$ from $\mathbf{x}_t$ using the posterior distribution $p(\mathbf{x}_{t-1}|\mathbf{x}_t, \mathbf{x}_0)$, which can be derived from the Bayes theorem using Eq. 4 and Eq. 5:

$$p(\mathbf{x}_{t-1}|\mathbf{x}_t, \mathbf{x}_0) = q(\mathbf{x}_t|\mathbf{x}_{t-1})\frac{q(\mathbf{x}_{t-1}|\mathbf{x}_0)}{q(\mathbf{x}_t|\mathbf{x}_0)} = \mathcal{N}(\mathbf{x}_{t-1}; \boldsymbol{\mu}_t(\mathbf{x}_t, \mathbf{x}_0), \sigma_t^2\mathbf{I}), \tag{6}$$

with the closed forms of mean $\boldsymbol{\mu}_t(\mathbf{x}_t, \mathbf{x}_0) = \frac{1}{\sqrt{\alpha_t}}\left(\mathbf{x}_t - \boldsymbol{\epsilon}\frac{1-\alpha_t}{\sqrt{1-\bar{\alpha}_t}}\right)$ and variance $\sigma_t^2 = \frac{1-\bar{\alpha}_{t-1}}{1-\bar{\alpha}_t}\beta_t$. $\boldsymbol{\epsilon}$ represents the noise in $\mathbf{x}_t$ and is the only uncertain variable during the reverse process. DDPM uses a neural network $\mathcal{Z}_{\boldsymbol{\theta}}$ to predict the noise $\boldsymbol{\epsilon}$ for each time-step $t$, i.e., $\boldsymbol{\epsilon}_t = \mathcal{Z}_{\boldsymbol{\theta}}(\mathbf{x}_t, t)$, where $\boldsymbol{\epsilon}_t$ denotes the estimation of $\boldsymbol{\epsilon}$ at time-step $t$. To train $\mathcal{Z}_{\boldsymbol{\theta}}$, DDPM randomly picks a clean image $\mathbf{x}_0$ from the dataset and samples a noise $\boldsymbol{\epsilon} \sim \mathcal{N}(0, \mathbf{I})$, then picks a random time-step $t$ and updates the network parameters $\boldsymbol{\theta}$ in $\mathcal{Z}_{\boldsymbol{\theta}}$ with the following gradient descent step (Ho et al., 2020):

$$\nabla_{\boldsymbol{\theta}}||\boldsymbol{\epsilon} - \mathcal{Z}_{\boldsymbol{\theta}}(\sqrt{\bar{\alpha}_t}\mathbf{x}_0 + \boldsymbol{\epsilon}\sqrt{1-\bar{\alpha}_t}, t)||_2^2. \tag{7}$$

By iteratively sampling $\mathbf{x}_{t-1}$ from $p(\mathbf{x}_{t-1}|\mathbf{x}_t, \mathbf{x}_0)$, DDPM can yield clean images $\mathbf{x}_0 \sim q(\mathbf{x})$ from random noises $\mathbf{x}_T \sim \mathcal{N}(\mathbf{0}, \mathbf{I})$, where $q(\mathbf{x})$ represents the image distribution in the training dataset.

### 2.2 RANGE-NULL SPACE DECOMPOSITION

For ease of derivation, we represent linear operators in matrix form and images in vector form. Note that our derivations hold for all linear operators. Given a linear operator $\mathbf{A} \in \mathbb{R}^{d \times D}$, its pseudo-inverse $\mathbf{A}^{\dagger} \in \mathbb{R}^{D \times d}$ satisfies $\mathbf{A}\mathbf{A}^{\dagger}\mathbf{A} \equiv \mathbf{A}$. There are many ways to solve the pseudo-inverse $\mathbf{A}^{\dagger}$, e.g., the Singular Value Decomposition (SVD) is often used to solve $\mathbf{A}^{\dagger}$ in matrix form, and the Fourier transform is often used to solve the convolutional form of $\mathbf{A}^{\dagger}$.

$\mathbf{A}$ and $\mathbf{A}^{\dagger}$ have some interesting properties. $\mathbf{A}^{\dagger}\mathbf{A}$ can be seen as the operator that projects samples $\mathbf{x} \in \mathbb{R}^{D \times 1}$ to the range-space of $\mathbf{A}$ because $\mathbf{A}\mathbf{A}^{\dagger}\mathbf{A}\mathbf{x} \equiv \mathbf{A}\mathbf{x}$. In contrast, $(\mathbf{I} - \mathbf{A}^{\dagger}\mathbf{A})$ can be seen as the operator that projects samples $\mathbf{x}$ to the null-space of $\mathbf{A}$ because $\mathbf{A}(\mathbf{I} - \mathbf{A}^{\dagger}\mathbf{A})\mathbf{x} \equiv \mathbf{0}$.

Interestingly, any sample $\mathbf{x}$ can be decomposed into two parts: one part is in the range-space of $\mathbf{A}$ and the other is in the null-space of $\mathbf{A}$, i.e.,

$$\mathbf{x} \equiv \mathbf{A}^{\dagger}\mathbf{A}\mathbf{x} + (\mathbf{I} - \mathbf{A}^{\dagger}\mathbf{A})\mathbf{x}. \tag{8}$$

This decomposition has profound significance for linear IR problems, which we will get to later.

## 3 METHOD

### 3.1 DENOISING DIFFUSION NULL-SPACE MODEL

**Null-Space Is All We Need.** We start with noise-free Image Restoration (IR) as below:

$$\mathbf{y} = \mathbf{A}\mathbf{x}, \tag{9}$$

where $\mathbf{x} \in \mathbb{R}^{D \times 1}$, $\mathbf{A} \in \mathbb{R}^{d \times D}$, and $\mathbf{y} \in \mathbb{R}^{d \times 1}$ denote the ground-truth (GT) image, the linear degradation operator, and the degraded image, respectively. Given an input $\mathbf{y}$, IR problems essentially aim to yield an image $\hat{\mathbf{x}} \in \mathbb{R}^{D \times 1}$ that conforms to the following two constraints:

$$Consistency: \quad \mathbf{A}\hat{\mathbf{x}} \equiv \mathbf{y}, \qquad Realness: \quad \hat{\mathbf{x}} \sim q(\mathbf{x}), \tag{10}$$

where $q(\mathbf{x})$ denotes the distribution of the GT images.

For the *Consistency* constraint, we can resort to range-null space decomposition. As discussed in Sec. 2.2, the GT image $\mathbf{x}$ can be decomposed as a range-space part $\mathbf{A}^\dagger \mathbf{A}\mathbf{x}$ and a null-space part $(\mathbf{I} - \mathbf{A}^\dagger \mathbf{A})\mathbf{x}$. Interestingly, we can find that the range-space part $\mathbf{A}^\dagger \mathbf{A}\mathbf{x}$ becomes exactly $\mathbf{y}$ after being operated by $\mathbf{A}$, while the null-space part $(\mathbf{I} - \mathbf{A}^\dagger \mathbf{A})\mathbf{x}$ becomes exactly $\mathbf{0}$ after being operated by $\mathbf{A}$, i.e., $\mathbf{A}\mathbf{x} \equiv \mathbf{A}\mathbf{A}^\dagger \mathbf{A}\mathbf{x} + \mathbf{A}(\mathbf{I} - \mathbf{A}^\dagger \mathbf{A})\mathbf{x} \equiv \mathbf{A}\mathbf{x} + \mathbf{0} \equiv \mathbf{y}$.

More interestingly, for a degraded image $\mathbf{y}$, we can directly construct a general solution $\hat{\mathbf{x}}$ that satisfies the *Consistency* constraint $\mathbf{A}\hat{\mathbf{x}} \equiv \mathbf{y}$, that is $\hat{\mathbf{x}} = \mathbf{A}^\dagger \mathbf{y} + (\mathbf{I} - \mathbf{A}^\dagger \mathbf{A})\bar{\mathbf{x}}$. Whatever $\bar{\mathbf{x}}$ is, it does not affect the *Consistency* at all. But $\bar{\mathbf{x}}$ determines whether $\hat{\mathbf{x}} \sim q(\mathbf{x})$. Then our goal is to find a proper $\bar{\mathbf{x}}$ that makes $\hat{\mathbf{x}} \sim q(\mathbf{x})$. We resort to diffusion models to generate the null-space $(\mathbf{I} - \mathbf{A}^\dagger \mathbf{A})\bar{\mathbf{x}}$ which is in harmony with the range-space $\mathbf{A}^\dagger \mathbf{y}$.

**Refine Null-Space Iteratively.** We know the reverse diffusion process iteratively samples $\mathbf{x}_{t-1}$ from $p(\mathbf{x}_{t-1}|\mathbf{x}_t, \mathbf{x}_0)$ to yield clean images $\mathbf{x}_0 \sim q(\mathbf{x})$ from random noises $\mathbf{x}_T \sim \mathcal{N}(\mathbf{0}, \mathbf{I})$. However, this process is completely random, and the intermediate state $\mathbf{x}_t$ is noisy. To yield clean intermediate states for range-null space decomposition, we reparameterize the mean $\boldsymbol{\mu}_t(\mathbf{x}_t, \mathbf{x}_0)$ and variance $\sigma_t^2$ of distribution $p(\mathbf{x}_{t-1}|\mathbf{x}_t, \mathbf{x}_0)$ as:

$$\boldsymbol{\mu}_t(\mathbf{x}_t, \mathbf{x}_0) = \frac{\sqrt{\bar{\alpha}_{t-1}}\beta_t}{1 - \bar{\alpha}_t}\mathbf{x}_0 + \frac{\sqrt{\alpha_t}(1 - \bar{\alpha}_{t-1})}{1 - \bar{\alpha}_t}\mathbf{x}_t, \quad \sigma_t^2 = \frac{1 - \bar{\alpha}_{t-1}}{1 - \bar{\alpha}_t}\beta_t, \tag{11}$$

where $\mathbf{x}_0$ is unknown, but we can reverse Eq. 5 to estimate a $\mathbf{x}_0$ from $\mathbf{x}_t$ and the predicted noise $\boldsymbol{\epsilon}_t = \mathcal{Z}_{\boldsymbol{\theta}}(\mathbf{x}_t, t)$. We denote the estimated $\mathbf{x}_0$ at time-step $t$ as $\mathbf{x}_{0|t}$, which can be formulated as:

$$\mathbf{x}_{0|t} = \frac{1}{\sqrt{\bar{\alpha}_t}}\left(\mathbf{x}_t - \mathcal{Z}_{\boldsymbol{\theta}}(\mathbf{x}_t, t)\sqrt{1 - \bar{\alpha}_t}\right). \tag{12}$$

Note that this formulation is equivalent to the original DDPM. We do this because it provides a "clean" image $\mathbf{x}_{0|t}$ (rather than noisy image $\mathbf{x}_t$). To finally yield a $\mathbf{x}_0$ satisfying $\mathbf{A}\mathbf{x}_0 \equiv \mathbf{y}$, we fix the range-space as $\mathbf{A}^\dagger \mathbf{y}$ and leave the null-space unchanged, yielding a rectified estimation $\hat{\mathbf{x}}_{0|t}$ as:

$$\hat{\mathbf{x}}_{0|t} = \mathbf{A}^\dagger \mathbf{y} + (\mathbf{I} - \mathbf{A}^\dagger \mathbf{A})\mathbf{x}_{0|t}. \tag{13}$$

Hence we use $\hat{\mathbf{x}}_{0|t}$ as the estimation of $\mathbf{x}_0$ in Eq. 11, thereby allowing only the null space to participate in the reverse diffusion process. Then we yield $\mathbf{x}_{t-1}$ by sampling from $p(\mathbf{x}_{t-1}|\mathbf{x}_t, \hat{\mathbf{x}}_{0|t})$:

$$\mathbf{x}_{t-1} = \frac{\sqrt{\bar{\alpha}_{t-1}}\beta_t}{1 - \bar{\alpha}_t}\hat{\mathbf{x}}_{0|t} + \frac{\sqrt{\alpha_t}(1 - \bar{\alpha}_{t-1})}{1 - \bar{\alpha}_t}\mathbf{x}_t + \sigma_t\boldsymbol{\epsilon}, \quad \boldsymbol{\epsilon} \sim \mathcal{N}(0, \mathbf{I}). \tag{14}$$

Roughly speaking, $\mathbf{x}_{t-1}$ is a noised version of $\hat{\mathbf{x}}_{0|t}$ and the added noise erases the disharmony between the range-space contents $\mathbf{A}^\dagger \mathbf{y}$ and the null-space contents $(\mathbf{I} - \mathbf{A}^\dagger \mathbf{A})\mathbf{x}_{0|t}$. Therefore, iteratively applying Eq. 12, Eq. 13, and Eq. 14 yields a final result $\mathbf{x}_0 \sim q(\mathbf{x})$. Note that all the rectified estimation $\hat{\mathbf{x}}_{0|t}$ conforms to *Consistency* due to the fact that

$$\mathbf{A}\hat{\mathbf{x}}_{0|t} \equiv \mathbf{A}\mathbf{A}^\dagger \mathbf{y} + \mathbf{A}(\mathbf{I} - \mathbf{A}^\dagger \mathbf{A})\mathbf{x}_{0|t} \equiv \mathbf{A}\mathbf{A}^\dagger \mathbf{A}\mathbf{x} + \mathbf{0} \equiv \mathbf{A}\mathbf{x} \equiv \mathbf{y}. \tag{15}$$

Considering $\mathbf{x}_0$ is equal to $\hat{\mathbf{x}}_{0|1}$, so the final result $\mathbf{x}_0$ also satisfies *Consistency*. We call the proposed method the Denoising Diffusion Null-Space Model (DDNM) because it utilizes the denoising diffusion model to fill up the null-space information.

| **Algorithm 1** Sampling of DDNM | **Algorithm 2** Sampling of DDNM$^+$ |
|---|---|
| 1: $\mathbf{x}_T \sim \mathcal{N}(\mathbf{0}, \mathbf{I})$ | 1: $\mathbf{x}_T \sim \mathcal{N}(\mathbf{0}, \mathbf{I})$ |
| 2: **for** $t = T, ..., 1$ **do** | 2: **for** $t = T, ..., 1$ **do** |
| | 3: $\quad L = \min\{T - t, l\}$ |
| | 4: $\quad \mathbf{x}_{t+L} \sim q(\mathbf{x}_{t+L}\|\mathbf{x}_j)$ |
| | 5: $\quad$ **for** $j = L, ..., 0$ **do** |
| 3: $\quad \mathbf{x}_{0\|t} = \frac{1}{\sqrt{\bar{\alpha}_t}}\left(\mathbf{x}_t - \mathcal{Z}_{\boldsymbol{\theta}}(\mathbf{x}_t, t)\sqrt{1 - \bar{\alpha}_t}\right)$ | 6: $\quad\quad \mathbf{x}_{0\|t+j} = \frac{1}{\sqrt{\bar{\alpha}_{t+j}}}\left(\mathbf{x}_{t+j} - \mathcal{Z}_{\boldsymbol{\theta}}(\mathbf{x}_{t+j}, t+j)\sqrt{1 - \bar{\alpha}_{t+j}}\right)$ |
| 4: $\quad \hat{\mathbf{x}}_{0\|t} = \mathbf{A}^\dagger\mathbf{y} + (\mathbf{I} - \mathbf{A}^\dagger\mathbf{A})\mathbf{x}_{0\|t}$ | 7: $\quad\quad \hat{\mathbf{x}}_{0\|t+j} = \mathbf{x}_{0\|t+j} - \boldsymbol{\Sigma}_{t+j}\mathbf{A}^\dagger(\mathbf{A}\mathbf{x}_{0\|t+j} - \mathbf{y})$ |
| 5: $\quad \mathbf{x}_{t-1} \sim p(\mathbf{x}_{t-1}\|\mathbf{x}_t, \hat{\mathbf{x}}_{0\|t})$ | 8: $\quad\quad \mathbf{x}_{t+j-1} \sim \hat{p}(\mathbf{x}_{t+j-1}\|\mathbf{x}_{t+j}, \hat{\mathbf{x}}_{0\|t+j})$ |
| 6: **return** $\mathbf{x}_0$ | 9: **return** $\mathbf{x}_0$ |

Figure 2: Illustration of (a) DDNM and (b) the time-travel trick.

Algo. 1 and Fig. 2(a) show the whole reverse diffusion process of DDNM. For ease of understanding, we visualize the intermediate results of DDNM in Appendix G. By using a denoising network $\mathcal{Z}_{\boldsymbol{\theta}}$ pre-trained for general generative purposes, DDNM can solve IR tasks with arbitrary forms of linear degradation operator $\mathbf{A}$. It does not need task-specific training or optimization and forms a zero-shot solution for diverse IR tasks.

It is worth noting that our method is compatible with most of the recent advances in diffusion models, e.g., DDNM can be deployed to score-based models (Song & Ermon, 2019; Song et al., 2020) or combined with DDIM (Song et al., 2021a) to accelerate the sampling speed.

## 3.2 EXAMPLES OF CONSTRUCTING $\mathbf{A}$ AND $\mathbf{A}^\dagger$

Typical IR tasks usually have simple forms of $\mathbf{A}$ and $\mathbf{A}^\dagger$, some of which are easy to construct by hand without resorting to complex Fourier transform or SVD. Here we introduce three practical examples. Inpainting is the simplest case, where $\mathbf{A}$ is the mask operator. Due to the unique property that $\mathbf{A}\mathbf{A}\mathbf{A} \equiv \mathbf{A}$, we can use $\mathbf{A}$ itself as $\mathbf{A}^\dagger$. For colorization, $\mathbf{A}$ can be a pixel-wise operator $\begin{bmatrix} \frac{1}{3} & \frac{1}{3} & \frac{1}{3} \end{bmatrix}$ that converts each RGB channel pixel $\begin{bmatrix} r & g & b \end{bmatrix}^\top$ into a grayscale value $\begin{bmatrix} \frac{r}{3} + \frac{g}{3} + \frac{b}{3} \end{bmatrix}$. It is easy to construct a pseudo-inverse $\mathbf{A}^\dagger = \begin{bmatrix} 1 & 1 & 1 \end{bmatrix}^\top$ that satisfies $\mathbf{A}\mathbf{A}^\dagger \equiv \mathbf{I}$. The same idea can be used for SR with scale $n$, where we can set $\mathbf{A} \in \mathbb{R}^{1 \times n^2}$ as the average-pooling operator $\begin{bmatrix} \frac{1}{n^2} & ... & \frac{1}{n^2} \end{bmatrix}$ that averages each patch into a single value. Similarly, we can construct its pseudo-inverse as $\mathbf{A}^\dagger \in \mathbb{R}^{n^2 \times 1} = \begin{bmatrix} 1 & ... & 1 \end{bmatrix}^\top$. We provide pytorch-like codes in Appendix E.

Considering $\mathbf{A}$ as a compound operation that consists of many sub-operations, i.e., $\mathbf{A} = \mathbf{A}_1...\mathbf{A}_n$, we may still yield its pseudo-inverse $\mathbf{A}^\dagger = \mathbf{A}_n^\dagger...\mathbf{A}_1^\dagger$. This provides a flexible solution for solving complex IR tasks, such as old photo restoration. Specifically, we can decompose the degradation of old photos as three parts, i.e., $\mathbf{A} = \mathbf{A}_1\mathbf{A}_2\mathbf{A}_3$, where $\mathbf{A}_3$ is the grayscale operator, $\mathbf{A}_2$ is the average-pooling operator with scale 4, and $\mathbf{A}_1$ is the mask operator defined by the damaged areas on the photo. Hence the pseudo-inverse is $\mathbf{A}^\dagger = \mathbf{A}_3^\dagger\mathbf{A}_2^\dagger\mathbf{A}_1^\dagger$. Our experiments show that these hand-designed operators work very well (Fig. 1(a,b,d)).

## 3.3 ENHANCED VERSION: DDNM$^+$

DDNM can solve noise-free IR tasks well but fails to handle noisy IR tasks and yields poor *Realness* in the face of some particular forms of $\mathbf{A}^\dagger$. To overcome these two limits, as described by **Algo.** 2, we propose an enhanced version, dubbed DDNM$^+$, by making the following two major extensions to DDNM to enable it to handle noisy situations and improve its restoration quality.

**Scaling Range-Space Correction to Support Noisy Image Restoration**   We consider noisy IR problems in the form of $\mathbf{y} = \mathbf{A}\mathbf{x} + \mathbf{n}$, where $\mathbf{n} \in \mathbb{R}^{d \times 1} \sim \mathcal{N}(\mathbf{0}, \sigma_{\mathbf{y}}^2\mathbf{I})$ represents the additive Gaussian noise and $\mathbf{A}\mathbf{x}$ represents the clean measurement. Applying DDNM directly yields

$$\hat{\mathbf{x}}_{0|t} = \mathbf{A}^\dagger\mathbf{y} + (\mathbf{I} - \mathbf{A}^\dagger\mathbf{A})\mathbf{x}_{0|t} = \mathbf{x}_{0|t} - \mathbf{A}^\dagger(\mathbf{A}\mathbf{x}_{0|t} - \mathbf{A}\mathbf{x}) + \mathbf{A}^\dagger\mathbf{n}, \tag{16}$$

where $\mathbf{A}^{\dagger}\mathbf{n} \in \mathbb{R}^{D \times 1}$ is the extra noise introduced into $\hat{\mathbf{x}}_{0|t}$ and will be further introduced into $\mathbf{x}_{t-1}$. $\mathbf{A}^{\dagger}(\mathbf{A}\mathbf{x}_{0|t} - \mathbf{A}\mathbf{x})$ is the correction for the range-space contents, which is the key to *Consistency*. To solve noisy image restoration, we propose to modify DDNM (on Eq. 13 and Eq. 14) as:

$$\hat{\mathbf{x}}_{0|t} = \mathbf{x}_{0|t} - \boldsymbol{\Sigma}_t \mathbf{A}^{\dagger}(\mathbf{A}\mathbf{x}_{0|t} - \mathbf{y}), \tag{17}$$

$$\hat{p}(\mathbf{x}_{t-1}|\mathbf{x}_t, \hat{\mathbf{x}}_{0|t}) = \mathcal{N}(\mathbf{x}_{t-1}; \boldsymbol{\mu}_t(\mathbf{x}_t, \hat{\mathbf{x}}_{0|t}), \boldsymbol{\Phi}_t \mathbf{I}). \tag{18}$$

$\boldsymbol{\Sigma}_t \in \mathbb{R}^{D \times D}$ is utilized to scale the range-space correction $\mathbf{A}^{\dagger}(\mathbf{A}\mathbf{x}_{0|t} - \mathbf{y})$ and $\boldsymbol{\Phi}_t \in \mathbb{R}^{D \times D}$ is used to scale the added noise $\sigma_t \boldsymbol{\epsilon}$ in $p(\mathbf{x}_{t-1}|\mathbf{x}_t, \hat{\mathbf{x}}_{0|t})$. The choice of $\boldsymbol{\Sigma}_t$ and $\boldsymbol{\Phi}_t$ follows two principles: (i) $\boldsymbol{\Sigma}_t$ and $\boldsymbol{\Phi}_t$ need to assure the total noise variance in $\mathbf{x}_{t-1}$ conforms to the definition in $q(\mathbf{x}_{t-1}|\mathbf{x}_0)$ (Eq. 5) so the total noise can be predicted by $\mathcal{Z}_{\boldsymbol{\theta}}$ and gets removed; (ii) $\boldsymbol{\Sigma}_t$ should be as close as possible to $\mathbf{I}$ to maximize the preservation of the range-space correction $\mathbf{A}^{\dagger}(\mathbf{A}\mathbf{x}_{0|t} - \mathbf{y})$ so as to maximize the *Consistency*. For SR and colorization defined in Sec.3.2, $\mathbf{A}^{\dagger}$ is copy operation. Thus $\mathbf{A}^{\dagger}\mathbf{n}$ can be approximated as a Gaussian noise $\mathcal{N}(\mathbf{0}, \sigma_{\mathbf{y}}^2 \mathbf{I})$, then $\boldsymbol{\Sigma}_t$ and $\boldsymbol{\Phi}_t$ can be simplified as $\boldsymbol{\Sigma}_t = \lambda_t \mathbf{I}$ and $\boldsymbol{\Phi}_t = \gamma_t \mathbf{I}$. Since $\mathbf{x}_{t-1} = \frac{\sqrt{\bar{\alpha}_{t-1}}\beta_t}{1-\bar{\alpha}_t}\hat{\mathbf{x}}_{0|t} + \frac{\sqrt{\alpha_t}(1-\bar{\alpha}_{t-1})}{1-\bar{\alpha}_t}\mathbf{x}_t + \sigma_t \boldsymbol{\epsilon}$, principle (i) is equivalent to: $(a_t \lambda_t \sigma_{\mathbf{y}})^2 + \gamma_t \equiv \sigma_t^2$ with $a_t$ denotes $\frac{\sqrt{\bar{\alpha}_{t-1}}\beta_t}{1-\bar{\alpha}_t}$. Considering principle (ii), we set:

$$\gamma_t = \sigma_t^2 - (a_t \lambda_t \sigma_{\mathbf{y}})^2, \quad \lambda_t = \begin{cases} 1, & \sigma_t \geq a_t \sigma_{\mathbf{y}} \\ \sigma_t / a_t \sigma_{\mathbf{y}}, & \sigma_t < a_t \sigma_{\mathbf{y}} \end{cases}. \tag{19}$$

In addition to the simplified version above, we also provide a more accurate version for general forms of $\mathbf{A}^{\dagger}$, where we set $\boldsymbol{\Sigma}_t = \mathbf{V}diag\{\lambda_{t1}, \ldots, \lambda_{tD}\}\mathbf{V}^{\top}$, $\boldsymbol{\Phi}_t = \mathbf{V}diag\{\gamma_{t1}, \ldots, \gamma_{tD}\}\mathbf{V}^{\top}$. $\mathbf{V}$ is derived from the SVD of the operator $\mathbf{A}(= \mathbf{U}\boldsymbol{\Sigma}\mathbf{V}^{\top})$. The calculation of $\lambda_{ti}$ and $\gamma_{ti}$ are presented in Appendix I. Note that the only hyperparameter that need manual setting is $\sigma_{\mathbf{y}}$.

We can also approximate non-Gaussian noise like Poisson, speckle, and real-world noise as Gaussian noise, thereby estimating a noise level $\sigma_{\mathbf{y}}$ and resorting to the same solution mentioned above.

**Time-Travel For Better Restoration Quality**   We find that DDNM yields inferior *Realness* when facing particular cases like SR with large-scale average-pooling downsampler, low sampling ratio compressed sensing(CS), and inpainting with a large mask. In these cases, the range-space contents $\mathbf{A}^{\dagger}\mathbf{y}$ is too local to guide the reverse diffusion process toward yielding a global harmony result.

Let us review Eq. 11. We can see that the mean value $\boldsymbol{\mu}_t(\mathbf{x}_t, \mathbf{x}_0)$ of the posterior distribution $p(\mathbf{x}_{t-1}|\mathbf{x}_t, \mathbf{x}_0)$ relies on accurate estimation of $\mathbf{x}_0$. DDNM uses $\hat{\mathbf{x}}_{0|t}$ as the estimation of $\mathbf{x}_0$ at time-step $t$, but if the range-space contents $\mathbf{A}^{\dagger}\mathbf{y}$ is too local or uneven, $\hat{\mathbf{x}}_{0|t}$ may have disharmonious null-space contents. How can we salvage the disharmony? Well, we can time travel back to change the past. Say we travel back to time-step $t + l$, we can yield the next state $\mathbf{x}_{t+l-1}$ using the "future" estimation $\hat{\mathbf{x}}_{0|t}$, which should be more accurate than $\hat{\mathbf{x}}_{0|t+l}$. By reparameterization, this operation is equivalent to sampling $\mathbf{x}_{t+l-1}$ from $q(\mathbf{x}_{t+l-1}|\mathbf{x}_{t-1})$. Similar to Lugmayr et al. (2022) that use a "back and forward" strategy for inpainting tasks, we propose a time-travel trick to improve global harmony for general IR tasks: For a chosen time-step $t$, we sample $\mathbf{x}_{t+l}$ from $q(\mathbf{x}_{t+l}|\mathbf{x}_t)$. Then we travel back to time-step $t + l$ and repeat normal DDNM sampling (Eq. 12, Eq. 13, and Eq. 14) until yielding $\mathbf{x}_{t-1}$. $l$ is actually the travel length. Fig. 2(b) illustrates the basic time-travel trick.

Intuitively, the time-travel trick produces a better "past", which in turn produces a better "future". For ease of use, we assign two extra hyperparameters: $s$ controls the interval of using the time-travel trick; $r$ determines the repeat times. The time-travel trick in Algo. 2 is with $s = 1$, $r = 1$. Fig. 4(b) and the right part in Tab. 4 demonstrate the improvements that the time-travel trick brings.

It is worth emphasizing that although Algo. 1 and Algo. 2 are derived based on DDPM, they can also be easily extended to other diffusion frameworks, such as DDIM (Song et al., 2021a). Obviously, DDNM$^+$ becomes exactly DDNM when setting $\boldsymbol{\Sigma}_t = \mathbf{I}$, $\boldsymbol{\Phi}_t = \sigma_t^2 \mathbf{I}$, and $l = 0$.

## 4   EXPERIMENTS

Our experiments consist of three parts. Firstly, we evaluate the performance of DDNM on five typical IR tasks and compare it with state-of-the-art zero-shot IR methods. Secondly, we experiment DDNM$^+$ on three typical IR tasks to verify its improvements against DDNM. Thirdly, we show that DDNM and DDNM$^+$ perform well on challenging real-world applications.

| ImageNet | 4× SR | Deblurring | Colorization | CS 25% | Inpainting |
|---|---|---|---|---|---|
| Method | PSNR↑/SSIM↑/FID↓ | PSNR↑/SSIM↑/FID↓ | Cons↓/FID↓ | PSNR↑/SSIM↑/FID↓ | PSNR↑/SSIM↑/FID↓ |
| $\mathbf{A}^\dagger\mathbf{y}$ | 24.26 / 0.684 / 134.4 | 18.56 / 0.6616 / 55.42 | 0.0 / 43.37 | 15.65 / 0.510 / 277.4 | 14.52 / 0.799 / 72.71 |
| DGP | 23.18 / 0.798 / 64.34 | N/A | - / 69.54 | N/A | N/A |
| ILVR | 27.40 / **0.870** / 43.66 | N/A | N/A | N/A | N/A |
| RePaint | N/A | N/A | N/A | N/A | 31.87 / **0.968** / 12.31 |
| DDRM | 27.38 / 0.869 / 43.15 | 43.01 / 0.992 / 1.48 | 260.4 / 36.56 | 19.95 / 0.704 / 97.99 | 31.73 / 0.966 / 4.82 |
| **DDNM**(ours) | **27.46 / 0.870/ 39.26** | **44.93 / 0.994 / 1.15** | **42.32 / 36.32** | **21.66 / 0.749 / 64.68** | **32.06 / 0.968 / 3.89** |
| CelebA-HQ | 4× SR | Deblurring | Colorization | CS 25% | Inpainting |
| Method | PSNR↑/SSIM↑/FID↓ | PSNR↑/SSIM↑/FID↓ | Cons↓/FID↓ | PSNR↑/SSIM↑/FID↓ | PSNR↑/SSIM↑/FID↓ |
| $\mathbf{A}^\dagger\mathbf{y}$ | 27.27 / 0.782 / 103.3 | 18.85 / 0.741 / 54.31 | 0.0 / 68.81 | 15.09 / 0.583 / 377.7 | 15.57 / 0.809 / 181.56 |
| PULSE | 22.74 / 0.623 / 40.33 | N/A | N/A | N/A | N/A |
| ILVR | 31.59 / **0.945** / 29.82 | N/A | N/A | N/A | N/A |
| RePaint | N/A | N/A | N/A | N/A | 35.20 / 0.981 /14.19 |
| DDRM | **31.63** / **0.945** / 31.04 | 43.07 / 0.993 / 6.24 | 455.9 / 31.26 | 24.86 / 0.876 / 46.77 | 34.79 / 0.978 /12.53 |
| **DDNM**(ours) | **31.63 / 0.945 / 22.27** | **46.72 / 0.996 / 1.41** | **26.25 / 26.44** | **27.56 / 0.909 / 28.80** | **35.64 / 0.982 /4.54** |

Table 1: Quantitative results of zero-shot IR methods on **ImageNet**(*top*) and **CelebA-HQ**(*bottom*), including five typical IR tasks. We mark N/A for those not applicable and **bold** the best scores.

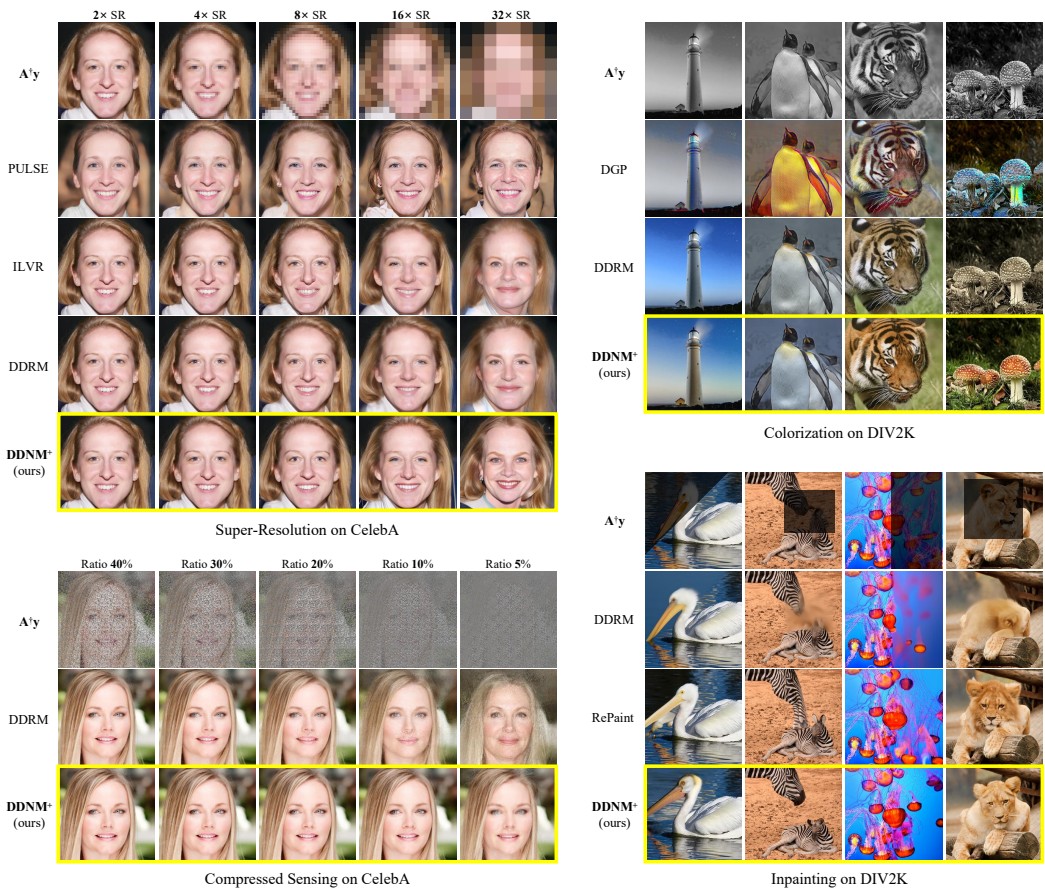

Figure 3: Qualitative results of zero-shot IR methods.

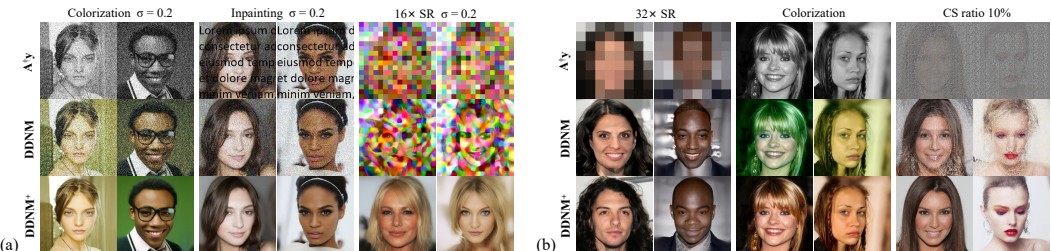

Figure 4: DDNM$^+$ improves (a) denoising performance and (b) restoration quality.

| CelebA-HQ | 16× SR $\sigma$=0.2 | C $\sigma$=0.2 | CS ratio=25% $\sigma$=0.2 | 32× SR | C | CS ratio=10% |
|---|---|---|---|---|---|---|
| Method | PSNR↑/SSIM↑/FID↓ | FID↓ | PSNR↑/SSIM↑/FID↓ | PSNR↑/SSIM↑/FID↓ | FID↓ | PSNR↑/SSIM↑/FID↓ |
| DDNM | 13.10 / 0.2387 / 281.45 | 216.74 | 17.89 / 0.4531 / 82.81 | 17.55 / 0.437 / 39.37 | 22.79 | 15.74/ 0.275 / 110.7 |
| DDNM$^+$ | **19.44 / 0.712 / 58.31** | **46.11** | **25.02 / 0.868 / 51.35** | **18.44 / 0.501 / 37.50** | **18.23** | **26.33 / 0.741 / 47.93** |

Table 2: Ablation study on denoising improvements (*left*) and the time-travel trick (*right*). C represents the colorization task. $\sigma$ denotes the noise variance on **y**.

## 4.1 EVALUATION ON DDNM

To evaluate the performance of DDNM, we compare DDNM with recent state-of-the-art zero-shot IR methods: DGP(Chen & Davies, 2020), Pulse(Menon et al., 2020), ILVR(Choi et al., 2021), RePaint(Lugmayr et al., 2022) and DDRM(Kawar et al., 2022). We experiment on five typical noise-free IR tasks, including 4× SR with bicubic downsampler, deblurring with Gaussian blur kernel, colorization with average grayscale operator, compressed sensing (CS) using Walsh-Hadamard sampling matrix with a 0.25 compression ratio, and inpainting with text masks. For each task, we use the same degradation operator for all methods. We choose ImageNet 1K and CelebA-HQ 1K datasets with image size 256×256 for validation. For ImageNet 1K, we use the 256×256 denoising network as $\mathcal{Z}_\theta$, which is pretrained on ImageNet by Dhariwal & Nichol (2021). For CelebA-HQ 1K, we use the 256×256 denoising network pretrained on CelebA-HQ by Lugmayr et al. (2022). For fair comparisons, we use the same pretrained denoising networks for ILVR, RePaint, DDRM, and DDNM. We use DDIM as the base sampling strategy with $\eta = 0.85$, 100 steps, without classifier guidance, for all diffusion-based methods. We choose PSNR, SSIM, and FID (Heusel et al., 2017) as the main metrics. Since PSNR and SSIM can not reflect the colorization performance, we use FID and the *Consistency* metric (calculated by $||\mathbf{Ax}_0 - \mathbf{y}||_1$ and denoted as *Cons*) for colorization.

Tab. 1 shows the quantitative results. For those tasks that are not supported, we mark them as "NA". We can see that DDNM far exceeds previous GAN prior based zero-shot IR methods (DGP, PULSE). Though with the same pretrained denoising models and sampling steps, DDNM achieves significantly better performance in both *Consistency* and *Realness* than ILVR, RePaint, and DDRM. Appendix J shows more quantitative comparisons and qualitative results.

## 4.2 EVALUATION ON DDNM$^+$

We evaluate the performance of DDNM$^+$ from two aspects: the denoising performance and the robustness in restoration quality.

**Denoising Performance.** We experiment DDNM$^+$ on three noisy IR tasks with $l = 0$, i.e., we disable the time-travel trick to only evaluate the denoising performance. Fig. 4(a) and the left part in Tab. 2 show the denoising improvements of DDNM$^+$ against DDNM. We can see that DDNM fully inherits the noise contained in **y**, while DDNM$^+$ decently removes the noise.

**Robustness in Restoration Quality.** We experiment DDNM$^+$ on three tasks that DDNM may yield inferior results, they are 32× SR, colorization, and compressed sensing (CS) using orthogonalized sampling matrix with a 10% compression ratio. For fair comparison, we set $T = 250$, $l = s = 20$, $r = 3$ for DDNM$^+$ while set $T = 1000$ for DDNM so that the total sampling steps and computational consumptions are roughly equal. Fig. 4(b) and the right part in Tab. 2 show the improvements of the time-travel trick. We can see that the time-travel trick significantly improves the overall performance, especially the *Realness* (measured by FID).

To the best of our knowledge, DDNM$^+$ is the first IR method that can robustly handle arbitrary scales of linear IR tasks. As is shown in Fig. 3, We compare DDNM$^+$ ($l = s = 10$, $r = 5$) with state-of-the-art zero-shot IR methods on diverse IR tasks. We also crop images from DIV2K dataset (Agustsson & Timofte, 2017) as the testset. The results show that DDNM$^+$ owns excellent robustness in dealing with diverse IR tasks, which is remarkable considering DDNM$^+$ as a zero-shot method. More experiments of DDNM/DDNM$^+$ can be found in Appendix A and B.

### 4.3 REAL-WORLD APPLICATIONS

Theoretically, we can use DDNM$^+$ to solve real-world IR task as long as we can construct an approximate linear degradation $\mathbf{A}$ and its pseudo-inverse $\mathbf{A}^\dagger$. Here we demonstrate two typical real-world applications using DDNM$^+$ with $l = s = 20$, $r = 3$: (1) **Real-World Noise.** We experiment DDNM$^+$ on real-world colorization with $\mathbf{A}$ and $\mathbf{A}^\dagger$ defined in Sec. 3.2. We set $\sigma_\mathbf{y}$ by observing the noise level of $\mathbf{y}$. The results are shown in Fig. 6, Fig. 7, and Fig. 1(c). (2) **Old Photo Restoration.** For old photos, we construct $\mathbf{A}$ and $\mathbf{A}^\dagger$ as described in Sec 3.2, where we manually draw a mask for damaged areas on the photo. The results are shown in Fig. 1(d), and Fig. 15.

## 5 RELATED WORK

### 5.1 DIFFUSION MODELS FOR IMAGE RESTORATION

Recent methods using diffusion models to solve image restoration can be roughly divided into two categories: supervised methods and zero-shot methods.

**Supervised Methods.** SR3 (Saharia et al., 2021) trains a conditional diffusion model for image super-resolution with synthetic image pairs as the training data. This pattern is further promoted to other IR tasks (Saharia et al., 2022). To solve image deblurring, Whang et al. (2022) uses a deterministic predictor to estimate the initial result and trains a diffusion model to predict the residual. However, these methods all need task-specific training and can not generalize to different degradation operators or different IR tasks.

**Zero-Shot Methods.** Song & Ermon (2019) first propose a zero-shot image inpainting solution by guiding the reverse diffusion process with the unmasked region. They further propose using gradient guidance to solve general inverse problems in a zero-shot fashion and apply this idea to medical imaging problems (Song et al., 2020; 2021b). ILVR (Choi et al., 2021) applies low-frequency guidance from a reference image to achieve reference-based image generation tasks. RePaint (Lugmayr et al., 2022) solves the inpainting problem by guiding the diffusion process with the unmasked region. DDRM (Kawar et al., 2022) uses SVD to decompose the degradation operators. However, SVD encounters a computational bottleneck when dealing with high-dimensional matrices. Actually, the core guidance function in ILVR (Choi et al., 2021), RePaint (Lugmayr et al., 2022) and DDRM (Kawar et al., 2022) can be seen as special cases of the range-null space decomposition used in DDNM, detailed analysis is in Appendix H.

### 5.2 RANGE-NULL SPACE DECOMPOSITION IN IMAGE INVERSE PROBLEMS

Schwab et al. (2019) first proposes using a DNN to learn the missing null-space contents in image inverse problems and provide detailed theory analysis. Chen & Davies (2020) proposes learning the range and null space respectively. Bahat & Michaeli (2020) achieves editable super-resolution via exploring the null-space contents. Wang et al. (2023) apply range-null space decomposition to existing GAN prior based SR methods to improve their performance and convergence speed.

## 6 CONCLUSION & DISCUSSION

This paper presents a unified framework for solving linear IR tasks in a zero-shot manner. We believe that our work demonstrates a promising new path for solving general IR tasks, which may also be instructive for general inverse problems. Theoretically, our framework can be easily extended to solve inverse problems of diverse data types, e.g., video, audio, and point cloud, as long as one can collect enough data to train a corresponding diffusion model. More discussions are in Appendix C.

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

## A  TIME & MEMORY CONSUMPTION

Our method has obvious advantages in time & memory consumption among recent zero-shot diffusion-based restoration methods (Kawar et al., 2022; Ho et al., 2022; Chung et al., 2022b;a). These methods are all based on basic diffusion models, the differences are how to bring the constraint $\mathbf{y} = \mathbf{A}\mathbf{x} + \mathbf{n}$ into the reverse diffusion process. We conclude our advantages as below:

- DDNM yields almost the same consumption as the original diffusion models.
- DDNM does not need any optimization toward minimizing $||\mathbf{y} - \mathbf{A}\mathbf{x}_{0|t}||$ since we directly yield the optimal solution by range-null space decomposition (Section 3.1) and precise range-space denoising (Section 3.3). We notice some recent works (Ho et al., 2022; Chung et al., 2022b;a) resort to such optimization, e.g., DPS (Chung et al., 2022a) uses $\mathbf{x}_{t-1} = \mathbf{x}_{t-1} - \zeta_t \nabla_{\mathbf{x}_t} ||\mathbf{y} - \mathbf{A}\mathbf{x}_{0|t}||_2^2$ to update $\mathbf{x}_{t-1}$; however, this involves costly gradient computation.
- Unlike DDRM (Kawar et al., 2022), our DDNM does not necessarily need SVD. As is presented in Section 3.2, we construct $\mathbf{A}$ and $\mathbf{A}^\dagger$ for colorization, inpainting, and super-resolution problems **by hand**, which bring negligible computation and memory consumption. In contrast, SVD-based methods suffer heavy cost on memory and computation if $\mathbf{A}$ has a high dimension (e.g., 128xSR, as shown below).

Experiments in Tab. 3 well support these claims.

| **ImageNet** | | | $4\times$ SR | | $64\times$ SR | | $128\times$ SR | |
| --- | --- | --- | --- | --- | --- | --- | --- | --- |
| Method | PSNR↑ | FID↓ | Time(s/image) | Memory(MB) | Time | Memory | Time | Memory |
| DDPM* | N/A | N/A | 11.9 | 5758 | 11.9 | 5758 | 11.9 | 5758 |
| DPS | 25.51 | 55.92 | 36.5 | 8112 | - | - | - | - |
| DDRM | **27.05** | 38.05 | 12.4 | 5788 | 36.4 | 5788 | 83.3 | 6792 |
| **DDNM** | 27.04 | **33.81** | **11.9** | **5728** | **11.9** | **5728** | **11.9** | **5728** |

Table 3: Comparisons on Time & Memory Consumption. We use the average-pooling downsampler, $4\times$ SR, 100 DDIM steps with $\eta$=0.85 and without classifier guidance, on a single 2080Ti GPU with batch size 1. For DPS, we set $\zeta_t$=$100\sqrt{\overline{\alpha}_{t-1}}$. *The DDPM here is tested on unconditional generation.

## B  COMPARING DDNM WITH SUPERVISED METHODS

Our method is superior to existing supervised IR methods (Zhang et al., 2021; Liang et al., 2021) in these ways:

- DDNM is zero-shot for diverse tasks, but supervised methods need to train separate models for each task.
- DDNM is robust to degradation modes, but supervised methods own poor generalized performance.
- DDNM yields significantly better performance on certain datasets and resolutions (e.g., ImageNet at 256x256).

These claims are well supported by experiments in Tab. 4.

| **ImageNet** | Bicubic, $\sigma_{\mathbf{y}}$=0 | | | Average-pooling, $\sigma_{\mathbf{y}}$=0 | | | Average-pooling, $\sigma_{\mathbf{y}}$=0.2 | | | Inference time |
| --- | --- | --- | --- | --- | --- | --- | --- | --- | --- | --- |
| Method | PSNR↑ | SSIM↑ | FID↓ | PSNR↑ | SSIM↑ | FID↓ | PSNR↑ | SSIM↑ | FID↓ | s/image |
| SwinIR-L | 21.21 | 0.7410 | 56.77 | 23.88 | 0.8010 | 54.93 | 18.39 | 0.5387 | 134.18 | 6.1 |
| BSRGAN | 21.46 | 0.7384 | 68.15 | 24.14 | 0.7948 | 67.70 | 14.06 | 0.3663 | 195.41 | **0.036** |
| DDNM | **27.46** | **0.8707** | **39.26** | **27.04** | **0.8651** | **33.81** | **22.67** | **0.7400** | **80.69** | 11.9 |

Table 4: Comparisons between DDNM and supervised SR methods. DDNM uses 100 DDIM steps with $\eta$=0.85 and without classifier guidance. We use the official SwinIR-L (Liang et al., 2021) and BSRGAN (Zhang et al., 2021) pretrained for SR tasks.

## C    LIMITATIONS

There remain many limitations that deserve further study.

- Though DDNM brings negligible extra cost on computations, it is still limited by the slow inference speed of existing diffusion models.
- DDNM needs explicit forms of the degradation operator, which may be challenging to acquire for some tasks. Approximations may work well, but not optimal.
- In theory, DDNM only supports linear operators. Though nonlinear operators may also have "pseudo-inverse", they may not conform to the distributive property, e.g., $\sin(a+b) \neq \sin(a) + \sin(b)$, so they may not have linearly separable null-space and range-space.
- DDNM inherits the randomness of diffusion models. This property benefits diversity but may yield undesirable results sometimes.
- The restoration capabilities of DDNM are limited by the performance of the pretrained denoiser, which is related to the network capacity and the training dataset. For example, existing diffusion models do not outperform StyleGANs (Karras et al., 2019; 2020; 2021) in synthesizing FFHQ/AFHQ images at 1024×1024 resolution.

## D    SOLVING REAL-WORLD DEGRADATION USING DDNM$^+$

DDNM+ can well handle real-world degradation, where the degradation operator A is unknown and non-linear and even contains non-Gaussian noise. We follow these observations:

- In theory, DDNM$^+$ is designed to solve IR tasks of diverse noise levels. As is shown in Fig. 5, DDNM$^+$ can well handle 4× SR even with a strong noise $\sigma_{\mathbf{y}}$=0.9.
- For real-world degraded images, the non-linear artifacts can generally be divided into **global** (e.g., the real-world noise in Fig. 1(c)) and **local** (e.g., the scratches in Fig. 1(d)).
- For **global** non-linear artifacts, we can set a proper $\sigma_{\mathbf{y}}$ to cover them. As is shown in Fig. 6, the input images $\mathbf{y}$ suffer JPEG-like unknown artifacts, but DDNM$^+$ can still remove them decently by setting a proper $\sigma_{\mathbf{y}}$.
- For **local** non-linear artifacts, we can directly draw a mask to cover them. Hence all we need is to construct $\mathbf{A} = \mathbf{A}_{color}\mathbf{A}_{mask}$ and set a proper $\sigma_{\mathbf{y}}$. We have proved $\mathbf{A}_{color}$ and $\mathbf{A}_{mask}$ and their pseudo-inverse can be easily constructed by hand. (maybe a $\mathbf{A}_{SR}$ is needed for resize when $\mathbf{y}$ is too blur)

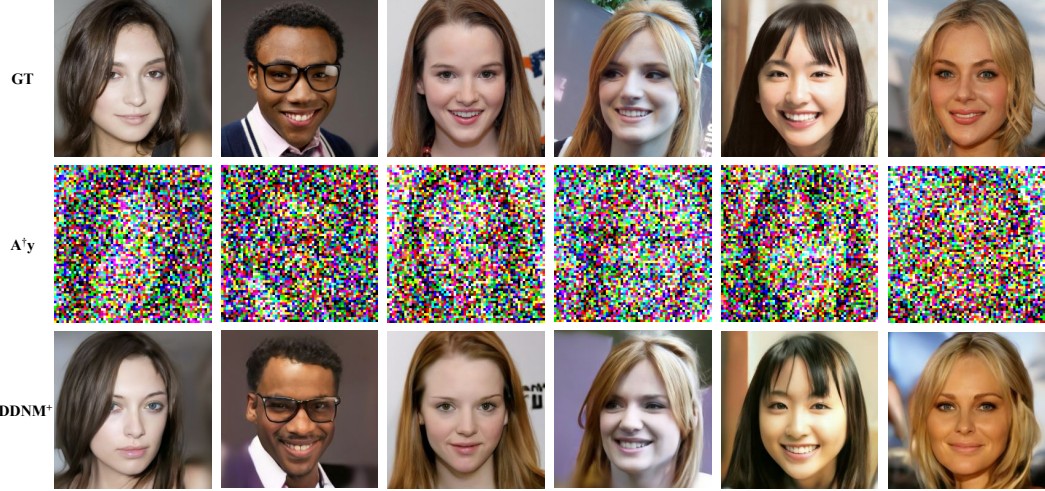

Figure 5: DDNM$^+$ can well handle 4× SR even with a strong noise $\sigma_{\mathbf{y}}$=0.9.

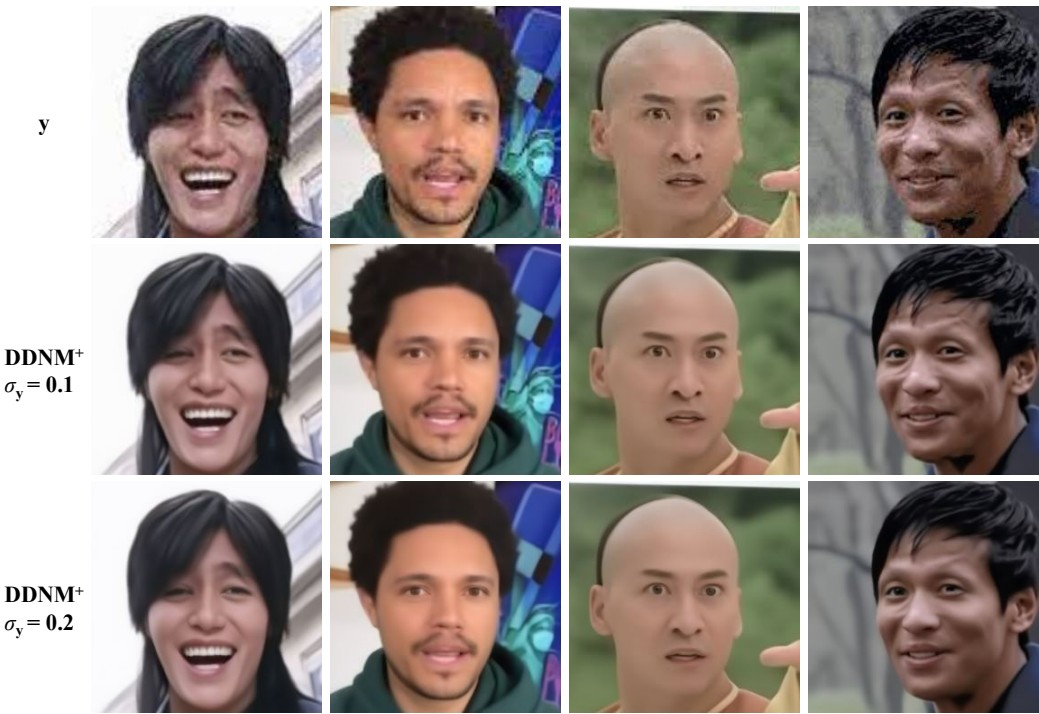

Figure 6: Solving JPEG-like artifacts using DDNM$^+$. Here we set $\mathbf{A} = \mathbf{I}$ to exert a pure denoising. $\mathbf{y}$ denotes the input degraded image. When we set $\sigma_{\mathbf{y}} = 0.1$, the artifacts are decently removed. When we set $\sigma_{\mathbf{y}} = 0.2$, the results become smoother but yield relatively poor identity consistency.

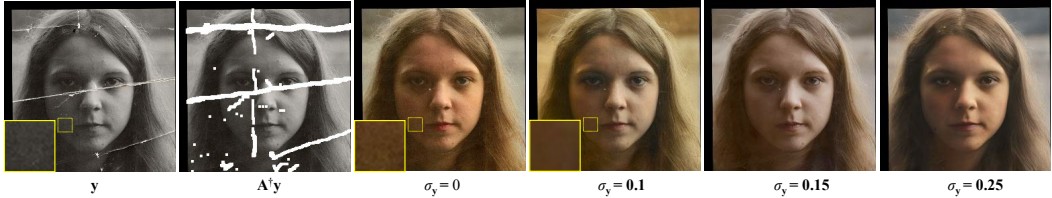

Figure 7: Old photo restoration. Zoom in for the best view. By setting $\sigma_{\mathbf{y}} = 0.1$, the noise is removed, and the identity is well preserved. When we set higher $\sigma_{\mathbf{y}} = 0.25$, the results becomes much smoother but yield relatively poor identity consistency.

In Fig. 7 we demonstrate an example. The input image $\mathbf{y}$ is a black-and-white photo with unknown noise and scratches. We first manually draw a mask $\mathbf{A_{mask}}$ to cover these scratches. Then we use a grayscale operator $\mathbf{A_{color}}$ to convert the image into grayscale. Definition of $\mathbf{A_{mask}}$ and $\mathbf{A_{color}}$ and their pseudo-inverse can be find in Sec. 3.2. Then we take $\mathbf{A} = \mathbf{A_{color}}\mathbf{A_{mask}}$ and $\mathbf{A}^{\dagger} = \mathbf{A_{mask}}^{\dagger}\mathbf{A_{color}}^{\dagger}$ for DDNM$^+$, and set a proper $\sigma_{\mathbf{y}}$. From the results in Fig. 7, we can see that when setting $\sigma_{\mathbf{y}} = 0$, the noise is fully inherited by the results. By setting $\sigma_{\mathbf{y}} = 0.1$, the noise is removed, and the identity is well preserved. When we set higher $\sigma_{\mathbf{y}} = 0.25$, the results becomes much smoother but yield relatively poor identity consistency.

The choice of $\sigma_{\mathbf{y}}$ is critical to achieve the best balance between realness and consistency. But for now we can only rely on manual estimates.

# E  PYTORCH-LIKE CODE IMPLEMENTATION

Here we provide a basic PyTorch-Like implementation of DDNM$^+$. Readers can quickly implement a basic DDNM$^+$ on their own projects by referencing Algo. 2 and Sec. 3.3 and the code below.

```python
def color2gray(x):
    coef=1/3
    x = x[:,0,:,:] * coef + x[:,1,:,:]*coef + x[:,2,:,:]*coef
    return x.repeat(1,3,1,1)

def gray2color(x):
    x = x[:,0,:,:]
    coef=1/3
    base = coef**2 + coef**2 + coef**2
    return th.stack((x*coef/base, x*coef/base, x*coef/base), 1)

def PatchUpsample(x, scale):
    n, c, h, w = x.shape
    x = torch.zeros(n,c,h,scale,w,scale) + x.view(n,c,h,1,w,1)
    return x.view(n,c,scale*h,scale*w)

# Implementation of A and its pseudo-inverse Ap

if IR_mode=="colorization":
    A = color2gray
    Ap = gray2color

elif IR_mode=="inpainting":
    A = lambda z: z*mask
    Ap = A

elif IR_mode=="super resolution":
    A = torch.nn.AdaptiveAvgPool2d((256//scale,256//scale))
    Ap = lambda z: PatchUpsample(z, scale)

elif IR_mode=="old photo restoration":
    A1 = lambda z: z*mask
    A1p = A1

    A2 = color2gray
    A2p = gray2color

    A3 = torch.nn.AdaptiveAvgPool2d((256//scale,256//scale))
    A3p = lambda z: PatchUpsample(z, scale)

    A = lambda z: A3(A2(A1(z)))
    Ap = lambda z: A1p(A2p(A3p(z)))

# Core Implementation of DDNM+, simplified denoising solution
# For more accurate denoising, please refer to Appendix I and the full source code.

def ddnmp_core(x0t, y, sigma_y, sigma_t, a_t):

    #Eq 19
    if sigma_t >= a_t*sigma_y:
        lambda_t = 1
        gamma_t = sigma_t**2 - (a_t*lambda_t*sigma_y)**2
    else:
        lambda_t = sigma_t/(a_t*sigma_y)
        gamma_t = 0

    #Eq 17
    x0t= x0t + lambda_t*Ap(y - A(x0t))

    return x0t, gamma_t
```

## F  DETAILS OF THE DEGRADATION OPERATORS

**Super Resolution (SR).**  For SR experiments in Tab. 1, we use the bicubic downsampler as the degradation operator to ensure fair comparisons. For other cases in this paper, we use the average-pooling downsampler as the degradation operator, which is easy to get the pseudo-inverse as described in Sec. 3.2. Fig. 8(a) and Fig. 8(b) show examples of the bicubic operation and the average-pooling operation.

**Inpainting.**  We use text masks, random pixel-wise masks, and hand-drawn masks for inpainting experiments. Fig.8(d) demonstrates examples of different masks.

**Deblurring.**  For deblurring experiments, We use three typical kernels to implement blurring operations, including Gaussian blur kernel, uniform blur kernel, and anisotropic blur kernel. For Gaussian blur, the kernel size is 5 and kernel width is 10; For uniform blur kernel, the kernel size is 9; For anisotropic blur kernel, the kernel size is 9 and the kernel widths of each axis are 20 and 1. Fig.8(c) demonstrates the effect of these kernels.

**Compressed Sensing (CS).**  For CS experiments, we choose two types of sampling matrices: one is based on the Walsh-Hadamard transformation, and the other is an orthogonalized random matrix applied to the original image block-wisely. For the Walsh-Hadamard sampling matrix, we choose 50% and 25% as the sampling ratio. For the orthogonalized sampling matrix, we choose ratios from 40% to 5%. Fig.8(e) and (f) demonstrate the effects of the Walsh-Hadamard sampling matrix and orthogonalized sampling matrix with different CS ratios.

**Colorization.**  For colorization, we choose the degradation matrix $\mathbf{A} = \begin{bmatrix} \frac{1}{3} & \frac{1}{3} & \frac{1}{3} \end{bmatrix}$ for each pixel as we described in Sec. 3.2. Fig.8(g) demonstrates the example of colorization degradation.

**Solve the Pseudo-Inverse Using SVD**  Considering we have a linear operator $\mathbf{A}$, we need to compute its pseudo-inverse $\mathbf{A}^{\dagger}$ to implement the algorithm of the proposed DDNM. For some simple degradation like inpainting, colorization, and SR based on average pooling, the pseudo-inverse $\mathbf{A}^{\dagger}$ can be constructed manually, which has been discussed in Sec. 3.2. For general cases, we can use the singular value decomposition (SVD) of $\mathbf{A}(= \mathbf{U\Sigma V}^{\top})$ to compute the pseudo-inverse $\mathbf{A}^{\dagger}(= \mathbf{V\Sigma}^{\dagger}\mathbf{U}^{\top})$ where $\mathbf{\Sigma}$ and $\mathbf{\Sigma}^{\dagger}$ have the following relationship:

$$\mathbf{\Sigma} = diag\{s_1, s_2, \cdots\}, \mathbf{\Sigma}^{\dagger} = diag\{d_1, d_2, \cdots\}, \tag{20}$$

$$d_i = \begin{cases} \frac{1}{s_i} & s_i \neq 0 \\ 0 & s_i = 0 \end{cases}, \tag{21}$$

where $s_i$ means the $i$-th singular value of $\mathbf{A}$ and $d_i$ means the $i$-th diagonal element of $\mathbf{\Sigma}^{\dagger}$.

## G  VISUALIZATION OF THE INTERMEDIATE RESULTS

In Fig. 9, we visualize the intermediate results of DDNM on $4\times$ SR, $16\times$ SR, and deblurring. Specifically, we show the noisy result $\mathbf{x}_t$, the clean estimation $\mathbf{x}_{0|t}$, and the rectified clean estimation $\hat{\mathbf{x}}_{0|t}$. The total diffusion step is 1000. From Fig. 9(a), we can see that due to the fixed range-space contents $\mathbf{A}^{\dagger}\mathbf{y}$, $\hat{\mathbf{x}}_{0|t}$ already owns meaningful contents in early stages while $\mathbf{x}_t$ and $\mathbf{x}_{0|t}$ contains limited information. But when $t = 0$, we can observe that $\mathbf{x}_{0|0}$ contains much more details than $\mathbf{A}^{\dagger}\mathbf{y}$. These details are precisely the null-space contents. We may notice a potential speed-up trick here. For example, we can replace $\mathbf{x}_{0|t=100}$ with $\mathbf{A}^{\dagger}\mathbf{y}$ and start DDNM directly from $t = 100$, which yields a 10 times faster sampling. We leave it to future work. From Fig. 9(b), we can see that the reverse diffusion process gradually restores images from low-frequency contours to high-frequency details.

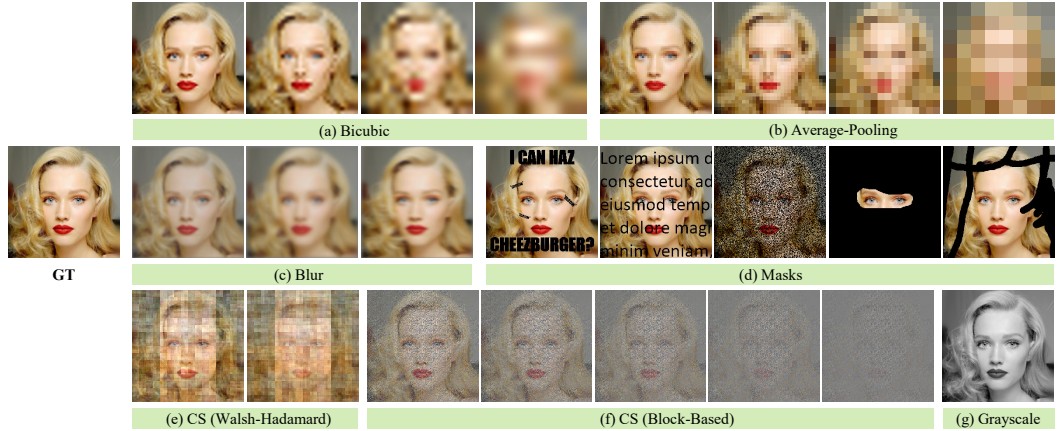

Figure 8: Visualization of different degradation operators. (a) Bicubic downsampler. The scale factors from left to right are ×4, ×8, ×16, ×32; (b) Average-pooling downsampler. The scale factors from left to right are ×4, ×8, ×16, ×32; (c) Blur operators. The type of kernels from left to right are Gaussian, uniform, and anisotropic; (d) Masks; (e) Walsh-Hadamard sampling matrix. The sampling ratios from left to right are 0.5 and 0.25; (f) Block-based sampling matrix. The sampling ratios from left to right are 0.4, 0.3, 0.2, 0.1, 0.05; (g) Grayscale operator.

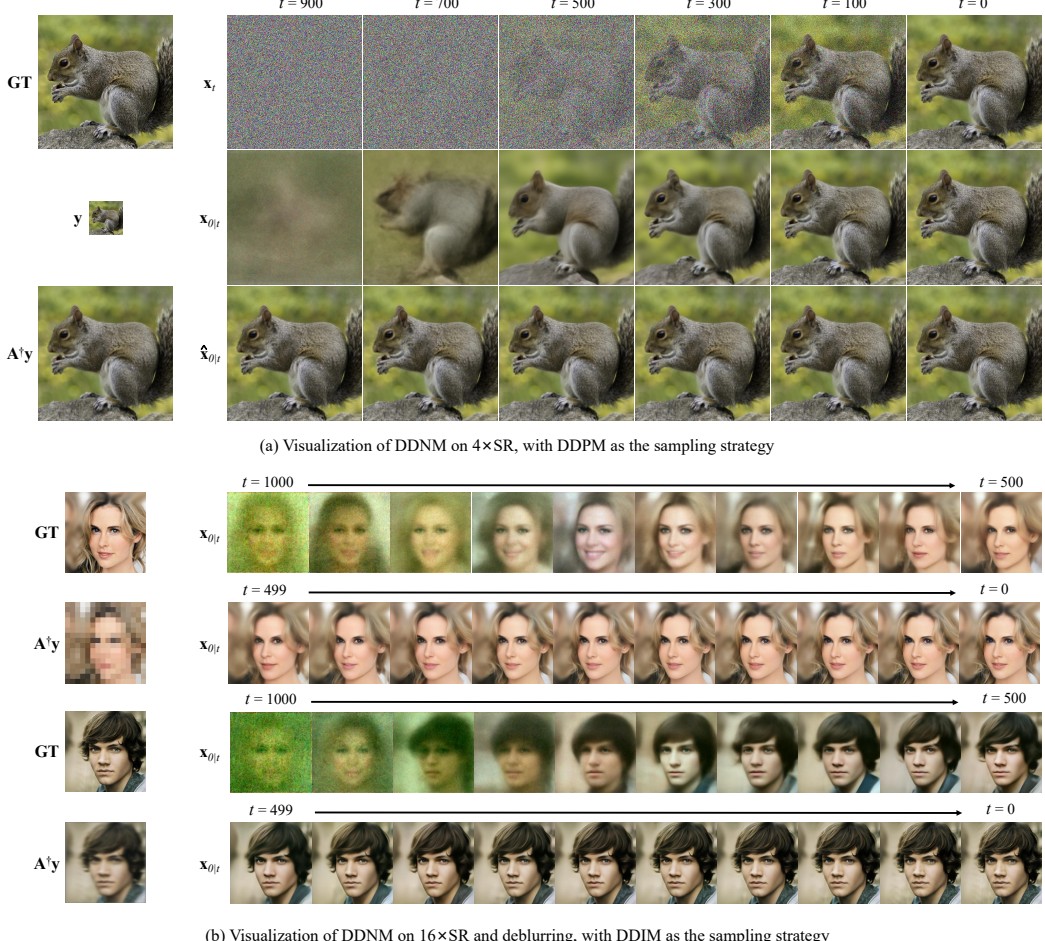

(a) Visualization of DDNM on 4×SR, with DDPM as the sampling strategy

(b) Visualization of DDNM on 16×SR and deblurring, with DDIM as the sampling strategy

Figure 9: Visualization of the intermediate results in DDNM. Zoom-in for the best view.

# H  COMPARING DDNM WITH RECENT DIFFUSION-BASED IR METHODS

Here we provide detailed comparison between DDNM and recent diffusion-based IR methods, including RePaint (Lugmayr et al., 2022), ILVR (Choi et al., 2021), DDRM (Kawar et al., 2022), SR3 (Saharia et al., 2021) and SDE (Song et al., 2020). For easier comparison, we rewrite their algorithms based on DDPM (Ho et al., 2020) and follow the characters used in DDNM. Algo. 3, Algo. 4 show the reverse diffusion process of DDPM and DDNM. We mark in blue those that are most distinct from DDNM. All the IR problems discussed here can be formulated as

$$\mathbf{y} = \mathbf{A}\mathbf{x} + \mathbf{n}, \tag{22}$$

where $\mathbf{y}$, $\mathbf{A}$, $\mathbf{x}$, $\mathbf{n}$ represents the degraded image, the degradation operator, the original image, and the additive noise, respectively.

## H.1  REPAINT AND ILVR.

RePaint (Lugmayr et al., 2022) solves noise-free image inpainting problems, where $\mathbf{n} = 0$ and $\mathbf{A}$ represents the mask operation. RePaint first create a noised version of the masked image $\mathbf{y}$

$$\mathbf{y}_{t-1} = \mathbf{A}(\sqrt{\bar{\alpha}_{t-1}}\mathbf{y} + \sqrt{1 - \bar{\alpha}_{t-1}}\boldsymbol{\epsilon}), \quad \boldsymbol{\epsilon} \sim \mathcal{N}(0, \mathbf{I}). \tag{23}$$

Then uses $\mathbf{y}_{t-1}$ to fill in the unmasked regions in $\mathbf{x}_{t-1}$:

$$\mathbf{x}_{t-1} = \mathbf{y}_{t-1} + (\mathbf{I} - \mathbf{A})\mathbf{x}_{t-1}, \tag{24}$$

Besides, RePaint applies an "back and forward" strategy to refine the results. Algo. 5 shows the algorithm of RePaint.

ILVR (Choi et al., 2021) focuses on reference-based image generation tasks, where $\mathbf{n} = 0$ and $\mathbf{A}$ represents a low-pass filter defined by $\mathbf{A} = \mathbf{A}_1\mathbf{A}_2$ ($\mathbf{A}_1$ is a bicubic upsampler and $\mathbf{A}_2$ is a bicubic downsampler). ILVR creates a noised version of the reference image $\mathbf{x}$ and uses the low-pass filter $\mathbf{A}$ to extract its low-frequency contents:

$$\mathbf{y}_{t-1} = \mathbf{A}(\sqrt{\bar{\alpha}_{t-1}}\mathbf{x} + \sqrt{1 - \bar{\alpha}_{t-1}}\boldsymbol{\epsilon}), \quad \boldsymbol{\epsilon} \sim \mathcal{N}(0, \mathbf{I}). \tag{25}$$

Then combines the high-frequency part of $\mathbf{x}_{t-1}$ with the low-frequency contents in $\mathbf{y}_{t-1}$:

$$\mathbf{x}_{t-1} = \mathbf{y}_{t-1} + (\mathbf{I} - \mathbf{A})\mathbf{x}_{t-1}, \tag{26}$$

Algo. 6 shows the algorithm of ILVR.

Essentially, RePaint and ILVR share the same formulations, with different definitions of the degradation operator $\mathbf{A}$. DDNM differs from RePaint and ILVR mainly in two parts:

(i) **Operating on Different Domains.** RePaint and ILVR all operate on the noisy $\mathbf{x}_t$ domain of diffusion models, which is inaccurate in range-space preservation during the reverse diffusion process. Instead, we directly operate on the noise-free $\mathbf{x}_{0|t}$ domain, which does not need extra process on $\mathbf{y}$ and is strictly derived from the theory and owns strict data consistency.

(ii) **As Special Cases.** Aside from the difference in operation domain, Eq. 24 of RePaint is essentially a special case of the range-null space decomposition. Considering $\mathbf{A}$ as a mask operator, it satisfies $\mathbf{A}\mathbf{A}\mathbf{A} = \mathbf{A}$, so we can use $\mathbf{A}$ itself as the pseudo-inverse $\mathbf{A}^\dagger$. Hence the range-null space decomposition becomes $\hat{\mathbf{x}} = \mathbf{A}^\dagger\mathbf{y} + (\mathbf{I} - \mathbf{A}^\dagger\mathbf{A})\bar{\mathbf{x}} = \mathbf{A}\mathbf{y} + (\mathbf{I} - \mathbf{A}\mathbf{A})\bar{\mathbf{x}} = \mathbf{y} + (\mathbf{I} - \mathbf{A})\bar{\mathbf{x}}$, which is exactly the same as Eq. 24. Similarly, Eq. 26 of ILVR can be seen as a special case of range-null space decomposition, which uses $\mathbf{I}$ as the approximation of $\mathbf{A}^\dagger$. Note that the final result $\mathbf{x}_0$ of RePaint satisfies *Consistency*, i.e., $\mathbf{A}\mathbf{x}_0 \equiv \mathbf{y}$, while ILVR does not because the pseudo-inverse $\mathbf{A}^\dagger$ they used is inaccurate.

**Algorithm 3** Reverse Diffusion Process of DDPM

**Require**: None
1: $\mathbf{x}_T \sim \mathcal{N}(\mathbf{0}, \mathbf{I})$.
2: **for** $t = T, ..., 1$ **do**
3:     $\boldsymbol{\epsilon} \sim \mathcal{N}(\mathbf{0}, \mathbf{I})$ if $t > 1$, else $\boldsymbol{\epsilon} = \mathbf{0}$.
4:     $\mathbf{x}_{t-1} = \frac{1}{\sqrt{\alpha_t}}\left(\mathbf{x}_t - \mathcal{Z}_{\boldsymbol{\theta}}(\mathbf{x}_t, t)\frac{\beta_t}{\sqrt{1-\bar{\alpha}_t}}\right) + \sigma_t \boldsymbol{\epsilon}$
5: **return** $\mathbf{x}_0$

---

**Algorithm 4** Reverse Diffusion Process of DDNM Based On DDPM

**Require**: The degraded image $\mathbf{y}$, the degradation operator $\mathbf{A}$ and its pseudo-inverse $\mathbf{A}^{\dagger}$
1: $\mathbf{x}_T \sim \mathcal{N}(\mathbf{0}, \mathbf{I})$.
2: **for** $t = T, ..., 1$ **do**
3:     $\boldsymbol{\epsilon} \sim \mathcal{N}(\mathbf{0}, \mathbf{I})$ if $t > 1$, else $\boldsymbol{\epsilon} = \mathbf{0}$.
4:     $\mathbf{x}_{0|t} = \frac{1}{\sqrt{\bar{\alpha}_t}}\left(\mathbf{x}_t - \mathcal{Z}_{\boldsymbol{\theta}}(\mathbf{x}_t, t)\sqrt{1-\bar{\alpha}_t}\right)$
5:     $\hat{\mathbf{x}}_{0|t} = \mathbf{x}_{0|t} - \mathbf{A}^{\dagger}(\mathbf{A}\mathbf{x}_{0|t} - \mathbf{y})$
6:     $\mathbf{x}_{t-1} = \frac{\sqrt{\bar{\alpha}_{t-1}}\beta_t}{1-\bar{\alpha}_t}\hat{\mathbf{x}}_{0|t} + \frac{\sqrt{\alpha_t}(1-\bar{\alpha}_{t-1})}{1-\bar{\alpha}_t}\mathbf{x}_t + \sigma_t\boldsymbol{\epsilon}$
7: **return** $\mathbf{x}_0$

---

**Algorithm 5** Reverse Diffusion Process of RePaint

**Require**: The masked image $\mathbf{y}$, the mask $\mathbf{A}$
1: $\mathbf{x}_T \sim \mathcal{N}(\mathbf{0}, \mathbf{I})$.
2: **for** $t = T, ..., 1$ **do**
3:     **for** $s = 1, ..., S_t$ **do**
4:       $\boldsymbol{\epsilon}_1, \boldsymbol{\epsilon}_2 \sim \mathcal{N}(\mathbf{0}, \mathbf{I})$ if $t > 1$, else $\boldsymbol{\epsilon}_1, \boldsymbol{\epsilon}_2 = \mathbf{0}$.
5:       $\mathbf{y}_{t-1} = \sqrt{\bar{\alpha}_{t-1}}\mathbf{y} + \sqrt{1-\bar{\alpha}_{t-1}}\boldsymbol{\epsilon}_1$
6:       $\mathbf{x}_{t-1} = \frac{1}{\sqrt{\alpha_t}}\left(\mathbf{x}_t - \mathcal{Z}_{\boldsymbol{\theta}}(\mathbf{x}_t, t)\frac{\beta_t}{\sqrt{1-\bar{\alpha}_t}}\right) + \sigma_t\boldsymbol{\epsilon}_2$
7:       $\mathbf{x}_{t-1} = \mathbf{y}_{t-1} + (\mathbf{I} - \mathbf{A})\mathbf{x}_{t-1}$
8:       **if** $t \neq 0$ and $s \neq S_t$ **then**
9:         $\mathbf{x}_t = \sqrt{1-\beta_t}\mathbf{x}_{t-1} + \sqrt{\beta_t}\boldsymbol{\epsilon}_2$
10: **return** $\mathbf{x}_0$

---

**Algorithm 6** Reverse Diffusion Process of ILVR

**Require**: The reference image $\mathbf{x}$, the low-pass filter $\mathbf{A}$
1: $\mathbf{x}_T \sim \mathcal{N}(\mathbf{0}, \mathbf{I})$.
2: **for** $t = T, ..., 1$ **do**
3:     $\boldsymbol{\epsilon}_1, \boldsymbol{\epsilon}_2 \sim \mathcal{N}(\mathbf{0}, \mathbf{I})$ if $t > 1$, else $\boldsymbol{\epsilon}_1, \boldsymbol{\epsilon}_2 = \mathbf{0}$.
4:     $\mathbf{y}_{t-1} = \mathbf{A}(\sqrt{\bar{\alpha}_{t-1}}\mathbf{x} + \sqrt{1-\bar{\alpha}_{t-1}}\boldsymbol{\epsilon}_1)$
5:     $\mathbf{x}_{t-1} = \frac{1}{\sqrt{\alpha_t}}\left(\mathbf{x}_t - \mathcal{Z}_{\boldsymbol{\theta}}(\mathbf{x}_t, t)\frac{\beta_t}{\sqrt{1-\bar{\alpha}_t}}\right) + \sigma_t\boldsymbol{\epsilon}_2$
6:     $\mathbf{x}_{t-1} = \mathbf{y}_{t-1} + (\mathbf{I} - \mathbf{A})\mathbf{x}_{t-1}$
7: **return** $\mathbf{x}_0$

**Algorithm 7** Reverse Diffusion Process of DDRM

**Require**: The degraded image $\mathbf{y}$ with noise level $\sigma_{\mathbf{y}}$, the operator $\mathbf{A} = \mathbf{U}\boldsymbol{\Sigma}\mathbf{V}^\top$, $\mathbf{A} \in \mathbb{R}^{d \times D}$

1: $\mathbf{x}_T \sim \mathcal{N}(\mathbf{0}, \mathbf{I})$.
2: $\bar{\mathbf{y}} = \boldsymbol{\Sigma}^\dagger \mathbf{U}^\top \mathbf{y}$
3: **for** $t = T, ..., 1$ **do**
4: $\quad \boldsymbol{\epsilon} \sim \mathcal{N}(0, \mathbf{I})$ if $t > 1$, else $\boldsymbol{\epsilon} = \mathbf{0}$.
5: $\quad \bar{\mathbf{x}}_{0|t} = \mathbf{V}^\top \frac{1}{\sqrt{\bar{\alpha}_t}} \left( \mathbf{x}_t - \mathcal{Z}_{\boldsymbol{\theta}}(\mathbf{x}_t, t)\sqrt{1 - \bar{\alpha}_t} \right)$
6: $\quad$ **for** $i = 1, ..., D$ **do**
7: $\qquad$ **if** $s_i = 0$ **then**
8: $\qquad\quad \bar{\mathbf{x}}_{t-1}^{(i)} = \bar{\mathbf{x}}_{0|t}^{(i)} + \sqrt{1 - \eta^2}\sigma_{t-1}\frac{\bar{\mathbf{x}}_t^{(i)} - \bar{\mathbf{x}}_{0|t}^{(i)}}{\sigma_t} + \eta\sigma_{t-1}\boldsymbol{\epsilon}^{(i)}$
9: $\qquad$ **else if** $\sigma_{t-1} < \frac{\sigma_{\mathbf{y}}}{s_i}$ **then**
10: $\qquad\quad \bar{\mathbf{x}}_{t-1}^{(i)} = \bar{\mathbf{x}}_{0|t}^{(i)} + \sqrt{1 - \eta^2}\sigma_{t-1}\frac{\bar{\mathbf{y}}^{(i)} - \bar{\mathbf{x}}_{0|t}^{(i)}}{\sigma_{\mathbf{y}}/s_i} + \eta\sigma_{t-1}\boldsymbol{\epsilon}^{(i)}$
11: $\qquad$ **else if** $\sigma_{t-1} \geq \frac{\sigma_{\mathbf{y}}}{s_i}$ **then**
12: $\qquad\quad \bar{\mathbf{x}}_{t-1}^{(i)} = \bar{\mathbf{y}}^{(i)} + \sqrt{\sigma_{t-1}^2 - \frac{\sigma_{\mathbf{y}}^2}{s_i^2}}\boldsymbol{\epsilon}^{(i)}$
13: $\quad \mathbf{x}_{t-1} = \mathbf{V}\bar{\mathbf{x}}_{t-1}$
14: **return** $\mathbf{x}_0$

## H.2 DDRM

The forward diffusion process defined by DDRM is

$$\mathbf{x}_t = \mathbf{x}_0 + \sigma_t\boldsymbol{\epsilon}, \quad \boldsymbol{\epsilon} \sim \mathcal{N}(\mathbf{0}, \mathbf{I}) \tag{27}$$

The original reverse diffusion process of DDRM is based on DDIM, which is

$$\mathbf{x}_{t-1} = \mathbf{x}_0 + \sqrt{1 - \eta^2}\sigma_{t-1}\frac{\mathbf{x}_t - \mathbf{x}_0}{\sigma_t} + \eta\sigma_{t-1}\boldsymbol{\epsilon} \tag{28}$$

For noisy linear inverse problem $\mathbf{y} = \mathbf{A}\mathbf{x} + \mathbf{n}$ where $\mathbf{n} \sim \mathcal{N}(0, \sigma_{\mathbf{y}}^2)$, DDRM first uses SVD to decompose $\mathbf{A}$ as $\mathbf{U}\boldsymbol{\Sigma}\mathbf{V}^\top$, then use $\bar{\mathbf{y}} = \boldsymbol{\Sigma}^\dagger\mathbf{U}^\top\mathbf{y}$ and $\bar{\mathbf{x}}_{0|t} = \mathbf{V}^\top\mathbf{x}_{0|t}$ for derivation. Each element in $\bar{\mathbf{y}}$ and $\bar{\mathbf{x}}_{0|t}$ corresponds to a singular value in $\boldsymbol{\Sigma}$(the nonexistent singular value is defined as 0), hence it is possible to modify $\mathbf{x}_{0|t}$ element-wisely according to each singular value. Then one can yield the final result $\mathbf{x}_0$ by $\mathbf{x}_0 = \mathbf{V}\bar{\mathbf{x}}_0$. Algo. 7 describes the whole reverse diffusion process of DDRM.

For noise-free($\sigma_{\mathbf{y}} = 0$) situation, the final result $\mathbf{x}_0$ of DDRM is essentially yielded through a special range-null space decomposition. Specifically, when $t = 0$ and $\sigma_{\mathbf{y}} = 0$, we can rewrite the formula of the $i$-th element of $\bar{\mathbf{x}}_0$ as:

$$\bar{\mathbf{x}}_0^{(i)} = \begin{cases} \bar{\mathbf{x}}_{0|1}^{(i)}, & s_i = 0 \\ \bar{\mathbf{y}}^{(i)}, & s_i \neq 0 \end{cases} \tag{29}$$

To simplify the representation, we define a diagonal matrix $\boldsymbol{\Sigma}_1$:

$$\boldsymbol{\Sigma}_1^{(i)} = \begin{cases} 0, & s_i = 0 \\ 1, & s_i \neq 0 \end{cases} \tag{30}$$

Then we can rewrite $\bar{\mathbf{x}}_0$ as

$$\bar{\mathbf{x}}_0 = \boldsymbol{\Sigma}_1\bar{\mathbf{y}} + (\mathbf{I} - \boldsymbol{\Sigma}_1)\bar{\mathbf{x}}_{0|1} \tag{31}$$

and yield the result $\mathbf{x}_0$ by left multiplying $\mathbf{V}$:

$$\mathbf{x}_0 = \mathbf{V}\bar{\mathbf{x}}_0 = \mathbf{V}\boldsymbol{\Sigma}_1\bar{\mathbf{y}} + \mathbf{V}(\mathbf{I} - \boldsymbol{\Sigma}_1)\bar{\mathbf{x}}_{0|1} \tag{32}$$

This result is essentially a special range-null space decomposition:

$$\begin{aligned}
\mathbf{x}_0 &= \mathbf{V}\boldsymbol{\Sigma}_1\bar{\mathbf{y}} + \mathbf{V}(\mathbf{I} - \boldsymbol{\Sigma}_1)\bar{\mathbf{x}}_{0|1} \\
&= \mathbf{V}\boldsymbol{\Sigma}_1\boldsymbol{\Sigma}^\dagger\mathbf{U}^\top\mathbf{y} + \mathbf{V}(\mathbf{I} - \boldsymbol{\Sigma}_1)\mathbf{V}^\top\mathbf{x}_{0|1} \\
&= \mathbf{V}\boldsymbol{\Sigma}^\dagger\mathbf{U}^\top\mathbf{y} + (\mathbf{I} - \mathbf{V}\boldsymbol{\Sigma}_1\mathbf{V}^\top)\mathbf{x}_{0|1} \\
&= \mathbf{A}^\dagger\mathbf{y} + (\mathbf{I} - \mathbf{A}^\dagger\mathbf{A})\mathbf{x}_{0|1}
\end{aligned} \tag{33}$$

Now we can clearly see that $\mathbf{V}\boldsymbol{\Sigma}_1\bar{\mathbf{y}} = \mathbf{A}^\dagger\mathbf{y}$ is the range-space part while $\mathbf{V}(\mathbf{I} - \boldsymbol{\Sigma}_1)\bar{\mathbf{x}}_{0|1} = (\mathbf{I} - \mathbf{A}^\dagger\mathbf{A})\mathbf{x}_{0|1}$ is the null-space part. However for our DDNM, $\mathbf{A}^\dagger$ can be any linear operator as long as it satisfies $\mathbf{A}\mathbf{A}^\dagger\mathbf{A} \equiv \mathbf{A}$, where $\mathbf{A}^\dagger = \mathbf{V}\boldsymbol{\Sigma}^\dagger\mathbf{U}^\top$ is a special case.

Due to the calculation needs of SVD, DDRM needs to convert the operator $\mathbf{A}$ into matrix form. However, common operations in computer vision are in the form of convolution, let alone $\mathbf{A}$ as a compound or high-dimension one. For example, DDRM is difficult to handle old photo restoration. Rather, our DDNM supports any linear forms of operator $\mathbf{A}$ and $\mathbf{A}^\dagger$, as long as $\mathbf{A}\mathbf{A}^\dagger\mathbf{A} = \mathbf{A}$ is satisfied. It is worth mentioning that there exist diverse ways of yielding the pseudo-inverse $\mathbf{A}^\dagger$, and SVD is just one of them. Besides, DDNM is more concise than DDRM in the formulation and performs better in noise-free IR tasks.

### H.3 OTHER DIFFUSION-BASED IR METHODS

SR3 (Saharia et al., 2021) is a task-specific super-resolution method which trains a denoiser with $\mathbf{y}$ as an additional input, i.e., $\mathcal{Z}_{\boldsymbol{\theta}}(\mathbf{x}_t, t, \mathbf{y})$. Then follow the similar reverse diffusion process in DDPM (Ho et al., 2020) to implement image super-resolution, as is shown in Algo. 8. SR3 needs to modify the network structures to support extra input $\mathbf{y}$ and needs paired data to train the conditional denoiser $\mathcal{Z}_{\boldsymbol{\theta}}(\mathbf{x}_t, t, \mathbf{y})$, while our DDNM is free from those burdens and is fully zero-shot for diverse IR tasks. Besides, DDNM can be also applied to SR3 to improve its performance. Specifically, we insert the core process of DDNM, the range-null space decomposition process, into SR3, yielding Algo.9. Results are demonstrated in Fig.10. We can see that the range-null space decomposition can improve the restoration quality by ensuring data consistency.

---

**Algorithm 8** Reverse Diffusion Process of SR3

**Require**: The degraded image $\mathbf{y}$
1: $\mathbf{x}_T \sim \mathcal{N}(\mathbf{0}, \mathbf{I})$.
2: **for** $t = T, ..., 1$ **do**
3: $\quad \boldsymbol{\epsilon} \sim \mathcal{N}(\mathbf{0}, \mathbf{I})$ if $t > 1$, else $\boldsymbol{\epsilon} = \mathbf{0}$.
4: $\quad \mathbf{x}_{t-1} = \frac{1}{\sqrt{\alpha_t}}\left(\mathbf{x}_t - \mathcal{Z}_{\boldsymbol{\theta}}(\mathbf{x}_t, t, \mathbf{y})\frac{\beta_t}{\sqrt{1-\bar{\alpha}_t}}\right) + \sigma_t\boldsymbol{\epsilon}$
5: **return** $\mathbf{x}_0$

---

**Algorithm 9** Reverse Diffusion Process of SR3+DDNM

**Require**: The degraded image $\mathbf{y}$
1: $\mathbf{x}_T \sim \mathcal{N}(\mathbf{0}, \mathbf{I})$.
2: **for** $t = T, ..., 1$ **do**
3: $\quad \boldsymbol{\epsilon} \sim \mathcal{N}(\mathbf{0}, \mathbf{I})$ if $t > 1$, else $\boldsymbol{\epsilon} = \mathbf{0}$.
4: $\quad \mathbf{x}_{0|t} = \frac{1}{\sqrt{\bar{\alpha}_t}}\left(\mathbf{x}_t - \mathcal{Z}_{\boldsymbol{\theta}}(\mathbf{x}_t, t, \mathbf{y})\sqrt{1 - \bar{\alpha}_t}\right)$
5: $\quad \hat{\mathbf{x}}_{0|t} = \mathbf{x}_{0|t} - \mathbf{A}^\dagger(\mathbf{A}\mathbf{x}_{0|t} - \mathbf{y})$
6: $\quad \mathbf{x}_{t-1} = \frac{\sqrt{\bar{\alpha}_{t-1}}\beta_t}{1-\bar{\alpha}_t}\hat{\mathbf{x}}_{0|t} + \frac{\sqrt{\alpha_t}(1-\bar{\alpha}_{t-1})}{1-\bar{\alpha}_t}\mathbf{x}_t + \sigma_t\boldsymbol{\epsilon}$
7: **return** $\mathbf{x}_0$

---

**Algorithm 10** Reverse Diffusion Process of SDE (conditional)

**Require**: The condition $\mathbf{y}$, the operator $\mathbf{A}$ and the rate $\lambda$
1: $\mathbf{x}_T \sim \mathcal{N}(\mathbf{0}, \mathbf{I})$.
2: **for** $t = T, ..., 1$ **do**
3: $\quad \boldsymbol{\epsilon}_1, \boldsymbol{\epsilon}_2 \sim \mathcal{N}(\mathbf{0}, \mathbf{I})$ if $t > 1$, else $\boldsymbol{\epsilon}_1, \boldsymbol{\epsilon}_2 = \mathbf{0}$.
4: $\quad \hat{\mathbf{x}}_t = \mathbf{x}_t + \lambda\nabla_{\mathbf{x}_t}f(\mathbf{A}\mathbf{x}_t, \mathbf{y})$
5: $\quad \mathbf{x}_{t-1} = \frac{1}{\sqrt{\alpha_t}}\left(\hat{\mathbf{x}}_t - \mathcal{Z}_{\boldsymbol{\theta}}(\hat{\mathbf{x}}_t, t)\frac{\beta_t}{\sqrt{1-\bar{\alpha}_t}}\right) + \sigma_t\boldsymbol{\epsilon}_2$
6: **return** $\mathbf{x}_0$

---

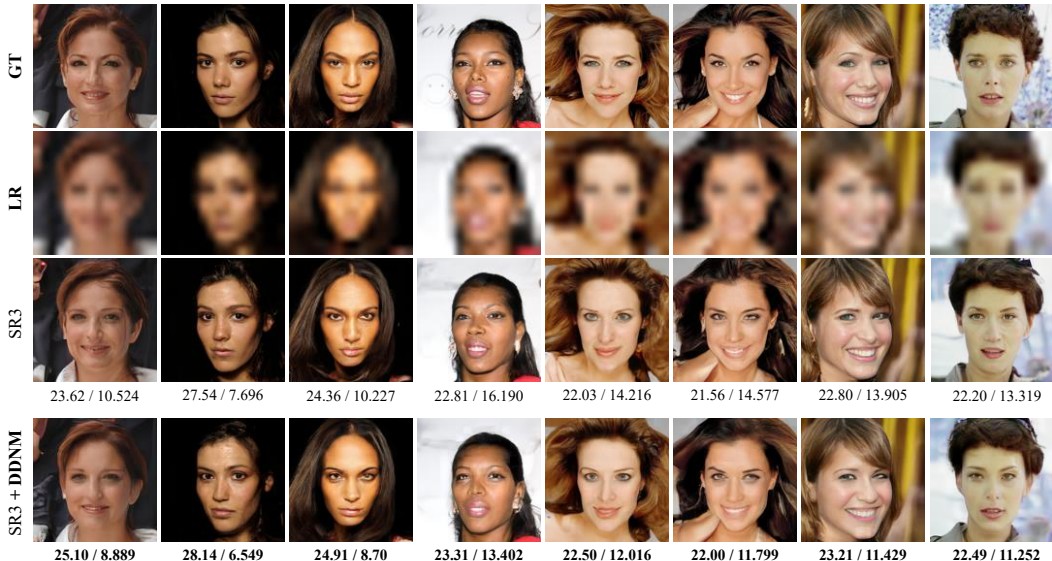

Figure 10: DDNM can be applied to SR3 to improve the restoration performance. Here we experiment on $8\times$ SR (from image size $16\times16$ to $128\times128$), the metrics are PSNR/*Consistency*.

Song et al. (2020) propose a conditional sampling strategy in diffusion models, which we abbreviate as SDE in this paper. Specifically, SDE optimize each latent variable $\mathbf{x}_t$ toward a specific condition $f(\mathbf{A}\mathbf{x}_t, \mathbf{y})$ and put the optimized $\mathbf{x}_t$ back to the original reverse diffusion process, as is shown in Algo. 10. $\mathbf{y}$ is the condition and $\mathbf{A}$ is an operator with $f(\cdot, \cdot)$ measures the distance between $\mathbf{A}\mathbf{x}_t$ and $\mathbf{y}$.

It is worth noting that DDNM is compatible with extra sources of constraints in the form of operation 5 in Algo. 10. For example, our results in Fig. 1 and Fig. 3 are generated using the diffusion model pretrained on ImageNet with classifier guidance.

## I  SOLVING NOISY IMAGE RESTORATION PRECISELY

For noisy tasks $\mathbf{y} = \mathbf{A}\mathbf{x} + \mathbf{n}, \mathbf{n} \sim \mathcal{N}(\mathbf{0}, \sigma_{\mathbf{y}}^2\mathbf{I})$, Sec. 3.3 provide a simple solution where $\mathbf{A}^\dagger\mathbf{n}$ is approximated as $\mathcal{N}(\mathbf{0}, \sigma_{\mathbf{y}}^2\mathbf{I})$. However, the precise distribution of $\mathbf{A}^\dagger\mathbf{n}$ is $\mathcal{N}(\mathbf{0}, \sigma_{\mathbf{y}}^2\mathbf{A}^\dagger(\mathbf{A}^\dagger)^T)$ where the covariance matrix is usually non-diagonal. To use similar principles in Eq. 19, we need to orthodiagonalize this matrix. Next, we conduct detailed derivations.

This solution involves the Singular Value Decomposition(SVD), which can decompose the degradation operator $\mathbf{A}$ and yield its pseudo-inverse $\mathbf{A}^\dagger$:

$$\mathbf{A} = \mathbf{U}\mathbf{\Sigma}\mathbf{V}^\top, \quad \mathbf{A}^\dagger = \mathbf{V}\mathbf{\Sigma}^\dagger\mathbf{U}^\top, \tag{34}$$

$$\mathbf{A} \in \mathbb{R}^{d\times D}, \mathbf{A}^\dagger \in \mathbb{R}^{D\times d}, \mathbf{U} \in \mathbb{R}^{d\times d}, \mathbf{V} \in \mathbb{R}^{D\times D}, \mathbf{\Sigma} \in \mathbb{R}^{d\times D}, \mathbf{\Sigma}^\dagger \in \mathbb{R}^{D\times d}, \tag{35}$$

$$\mathbf{\Sigma} = diag\{s_1, s_2, \cdots, s_d\}, \quad \mathbf{\Sigma}^{(i)} = s_i, \quad \mathbf{\Sigma}^{\dagger(i)} = \begin{cases} \frac{1}{s_i}, & s_i \neq 0, \\ 0, & s_i = 0 \end{cases}, \tag{36}$$

To find out how much noise has been introduced into $\hat{\mathbf{x}}_{0|t}$, we first rewrite Eq. 17 as:

$$\hat{\mathbf{x}}_{0|t} = \mathbf{x}_{0|t} - \mathbf{\Sigma_t}\mathbf{A}^\dagger(\mathbf{A}\mathbf{x}_{0|t} - \mathbf{A}\mathbf{x} - \mathbf{n}), \tag{37}$$

where $\mathbf{A}\mathbf{x}$ represents the clean measurements before adding noise. $\mathbf{\Sigma_t} = \mathbf{V}\Lambda_t\mathbf{V}^\top$ is the scaling matrix with $\Lambda_t = diag\{\lambda_{t1}, \lambda_{t2}, \cdots, \lambda_{tD}\}$. Then we can rewrite the additive noise $\mathbf{n}$ as $\sigma_{\mathbf{y}}\boldsymbol{\epsilon}_{\mathbf{n}}$ where $\boldsymbol{\epsilon}_{\mathbf{n}} \sim \mathcal{N}(\mathbf{0}, \mathbf{I})$. Now Eq. 37 becomes

$$\hat{\mathbf{x}}_{0|t} = \mathbf{x}_{0|t} - \mathbf{\Sigma}_t\mathbf{A}^\dagger(\mathbf{A}\mathbf{x}_{0|t} - \mathbf{A}\mathbf{x}) + \sigma_{\mathbf{y}}\mathbf{V}\Lambda_t\mathbf{V}^\top\mathbf{A}^\dagger\boldsymbol{\epsilon}_{\mathbf{n}}, \tag{38}$$

where $\mathbf{x}_{0|t} - \mathbf{\Sigma}_t \mathbf{A}^\dagger (\mathbf{A}\mathbf{x}_{0|t} - \mathbf{A}\mathbf{x})$ denotes the clean part of $\hat{\mathbf{x}}_{0|t}$ (written as $\hat{\mathbf{x}}_{0|t}^c$). It is clear that the noise introduced into $\hat{\mathbf{x}}_{0|t}$ is $\sigma_{\mathbf{y}} \mathbf{V} \Lambda_t \mathbf{V}^\top \mathbf{A}^\dagger \boldsymbol{\epsilon}_{\mathbf{n}}$. The handling of the introduced noise depends on the sampling strategy we used. We will discuss the solution for DDPM and DDIM, respectively.

**The Situation in DDPM.** When using DDPM as the sampling strategy, we yield $\mathbf{x}_{t-1}$ by sampling from $p(\mathbf{x}_{t-1}|\mathbf{x}_t, \mathbf{x}_0) = \mathcal{N}(\mathbf{x}_{t-1}; \mu_t(\mathbf{x}_t, \mathbf{x}_0), \sigma_t^2 \mathbf{I})$, i.e.,

$$\mathbf{x}_{t-1} = \frac{\sqrt{\bar{\alpha}_{t-1}}\beta_t}{1-\bar{\alpha}_t}\hat{\mathbf{x}}_{0|t} + \frac{\sqrt{\alpha_t}(1-\bar{\alpha}_{t-1})}{1-\bar{\alpha}_t}\mathbf{x}_t + \sigma_t \boldsymbol{\epsilon}, \quad \boldsymbol{\epsilon} \sim \mathcal{N}(\mathbf{0}, \mathbf{I}), \tag{39}$$

Considering the introduced noise, we change $\sigma_t \boldsymbol{\epsilon}$ to ensure the entire noise level not exceed $\mathcal{N}(\mathbf{0}, \sigma_t^2 \mathbf{I})$. Hence we construct a new noise $\boldsymbol{\epsilon}_{new} \sim \mathcal{N}(\mathbf{0}, \mathbf{\Phi}_t \mathbf{I})$. Then the Eq. 39 becomes

$$\mathbf{x}_{t-1} = \frac{\sqrt{\bar{\alpha}_{t-1}}\beta_t}{1-\bar{\alpha}_t}\hat{\mathbf{x}}_{0|t}^c + \frac{\sqrt{\alpha_t}(1-\bar{\alpha}_{t-1})}{1-\bar{\alpha}_t}\mathbf{x}_t + \boldsymbol{\epsilon}_{intro} + \boldsymbol{\epsilon}_{new}, \tag{40}$$

$$\boldsymbol{\epsilon}_{intro} = \frac{\sqrt{\bar{\alpha}_{t-1}}\beta_t}{1-\bar{\alpha}_t}\sigma_{\mathbf{y}}\mathbf{V}\Lambda_t\mathbf{V}^\top\mathbf{A}^\dagger\boldsymbol{\epsilon}_{\mathbf{n}}, \tag{41}$$

$$\boldsymbol{\epsilon}_{intro} + \boldsymbol{\epsilon}_{new} \sim \mathcal{N}(\mathbf{0}, \sigma_t^2 \mathbf{I}). \tag{42}$$

$\boldsymbol{\epsilon}_{intro}$ denotes the introduced noise, which can be further written as

$$\boldsymbol{\epsilon}_{intro} = \frac{\sqrt{\bar{\alpha}_{t-1}}\beta_t}{1-\bar{\alpha}_t}\sigma_{\mathbf{y}}\mathbf{V}\Lambda_t\mathbf{V}^\top\mathbf{A}^\dagger\boldsymbol{\epsilon}_{\mathbf{n}} \tag{43}$$

$$\sim \mathcal{N}(\mathbf{0}, (\frac{\sqrt{\bar{\alpha}_{t-1}}\beta_t}{1-\bar{\alpha}_t})^2\sigma_{\mathbf{y}}^2(\mathbf{V}\Lambda_t\mathbf{V}^\top\mathbf{A}^\dagger)\mathbf{I}(\mathbf{V}\Lambda_t\mathbf{V}^\top\mathbf{A}^\dagger)^\top) \tag{44}$$

$$\sim \mathcal{N}(\mathbf{0}, (\frac{\sqrt{\bar{\alpha}_{t-1}}\beta_t}{1-\bar{\alpha}_t})^2\sigma_{\mathbf{y}}^2\mathbf{V}\Lambda_t\mathbf{V}^\top\mathbf{A}^\dagger(\mathbf{A}^\dagger)^\top\mathbf{V}\Lambda_t\mathbf{V}^\top) \tag{45}$$

$$\sim \mathcal{N}(\mathbf{0}, (\frac{\sqrt{\bar{\alpha}_{t-1}}\beta_t}{1-\bar{\alpha}_t})^2\sigma_{\mathbf{y}}^2\mathbf{V}\Lambda_t\mathbf{V}^\top\mathbf{V}\mathbf{\Sigma}^\dagger\mathbf{U}^\top\mathbf{U}(\mathbf{\Sigma}^\dagger)^\top\mathbf{V}^\top\mathbf{V}\Lambda_t\mathbf{V}^\top) \tag{46}$$

$$\sim \mathcal{N}(\mathbf{0}, (\frac{\sqrt{\bar{\alpha}_{t-1}}\beta_t}{1-\bar{\alpha}_t})^2\sigma_{\mathbf{y}}^2\mathbf{V}\Lambda_t\mathbf{\Sigma}^\dagger(\mathbf{\Sigma}^\dagger)^\top\Lambda_t\mathbf{V}^\top) \tag{47}$$

The variance matrix of $\boldsymbol{\epsilon}_{intro}$ can be simplified as $\mathbf{V}\mathbf{D}_t\mathbf{V}^\top$, with $\mathbf{D}_t = diag\{d_{t1}, d_{t2}, \cdots, d_{tD}\}$:

$$\boldsymbol{\epsilon}_{intro} \sim \mathcal{N}(\mathbf{0}, \mathbf{V}\mathbf{D}_t\mathbf{V}^\top), \quad d_{ti} = \begin{cases} \frac{(\frac{\sqrt{\bar{\alpha}_{t-1}}\beta_t}{1-\bar{\alpha}_t})^2\sigma_{\mathbf{y}}^2\lambda_{ti}^2}{s_i^2}, & s_i \neq 0, \\ 0, & s_i = 0 \end{cases}, \tag{48}$$

To construct $\boldsymbol{\epsilon}_{new}$, we define a new diagonal matrix $\mathbf{\Gamma}_t(= diag\{\gamma_{t1}, \gamma_{t2}, \cdots, \gamma_{tD}\})$:

$$\mathbf{\Gamma}_t = \sigma_t^2\mathbf{I} - \mathbf{D}_t, \quad \gamma_{ti} = \begin{cases} \sigma_t^2 - \frac{(\frac{\sqrt{\bar{\alpha}_{t-1}}\beta_t}{1-\bar{\alpha}_t})^2\sigma_{\mathbf{y}}^2\lambda_{ti}^2}{s_i^2}, & s_i \neq 0, \\ \sigma_t^2, & s_i = 0 \end{cases}, \tag{49}$$

Now we can yield $\boldsymbol{\epsilon}_{new}$ by sampling from $\mathcal{N}(\mathbf{0}, \mathbf{V}\mathbf{\Gamma}_t\mathbf{V}^\top)$ to ensure that $\boldsymbol{\epsilon}_{intro} + \boldsymbol{\epsilon}_{new} \sim \mathcal{N}(\mathbf{0}, \mathbf{V}(\mathbf{D}_t + \mathbf{\Gamma}_t)\mathbf{V}^\top) = \mathcal{N}(\mathbf{0}, \sigma_t^2\mathbf{I})$. An easier implementation method is firstly sampling $\boldsymbol{\epsilon}_{temp}$ from $\mathcal{N}(\mathbf{0}, \mathbf{\Gamma}_t)$ and finally get $\boldsymbol{\epsilon}_{new} = \mathbf{V}\boldsymbol{\epsilon}_{temp}$. From Eq. 49, we also observe that $\lambda_{ti}$ guarantees the noise level of the introduced noise do not exceed the pre-defined noise level $\sigma_t$ so that we can get the formula of $\lambda_{ti}$ in $\mathbf{\Sigma}_t(= \mathbf{V}\Lambda_t\mathbf{V}^\top, \Lambda_t = diag\{\lambda_{t1}, \lambda_{t2}, \cdots, \lambda_{tD}\})$:

$$\lambda_{ti} = \begin{cases} 1, & \sigma_t \geq \frac{(\frac{\sqrt{\bar{\alpha}_{t-1}}\beta_t}{1-\bar{\alpha}_t})\sigma_{\mathbf{y}}}{s_i} \\ \frac{\sigma_t s_i}{(\frac{\sqrt{\bar{\alpha}_{t-1}}\beta_t}{1-\bar{\alpha}_t})\sigma_{\mathbf{y}}}, & \sigma_t < \frac{(\frac{\sqrt{\bar{\alpha}_{t-1}}\beta_t}{1-\bar{\alpha}_t})\sigma_{\mathbf{y}}}{s_i} \\ 1, & s_i = 0 \end{cases}, \tag{50}$$

**The Situation in DDIM.** When using DDIM as the sampling strategy, the process of getting $\mathbf{x}_{t-1}$ from $\mathbf{x}_t$ becomes:

$$\mathbf{x}_{t-1} = \sqrt{\bar{\alpha}_{t-1}}\hat{\mathbf{x}}_{0|t} + \sigma_t\sqrt{1-\eta^2}\mathcal{Z}_{\boldsymbol{\theta}}(\mathbf{x}_t, t) + \sigma_t\eta\boldsymbol{\epsilon}, \quad \boldsymbol{\epsilon} \sim \mathcal{N}(\mathbf{0}, \mathbf{I}), \tag{51}$$

where $\sigma_t = \sqrt{1 - \bar{\alpha}_{t-1}}$ is the noise level of the $t$-th time-step, $\mathcal{Z}_{\boldsymbol{\theta}}$ is the denoiser which estimates the additive noise from $\mathbf{x}_t$ and $\eta$ control the randomness of this sampling process. Considering the noise part is subject to a normal distribution, that is, $\sigma_t \sqrt{1 - \eta^2} \mathcal{Z}_{\boldsymbol{\theta}}(\mathbf{x}_t, t) + \sigma_t \eta \boldsymbol{\epsilon} \sim \mathcal{N}(\mathbf{0}, \sigma_t^2 \mathbf{I})$, so that the equation can be rewritten as

$$\mathbf{x}_{t-1} = \sqrt{\bar{\alpha}_{t-1}} \hat{\mathbf{x}}_{0|t} + \boldsymbol{\epsilon}_{orig}, \quad \boldsymbol{\epsilon}_{orig} \sim \mathcal{N}(\mathbf{0}, \sigma_t^2 \mathbf{I}) \tag{52}$$

Considering the introduced noise, we change $\boldsymbol{\epsilon}_{orig}$ to ensure the entire noise level not exceed $\mathcal{N}(\mathbf{0}, \sigma_t^2 \mathbf{I})$. Hence we construct a new noise term $\boldsymbol{\epsilon}_{new} \sim \mathcal{N}(\mathbf{0}, \boldsymbol{\Phi}_t \mathbf{I})$:

$$\mathbf{x}_{t-1} = \sqrt{\bar{\alpha}_{t-1}} \hat{\mathbf{x}}_{0|t}^c + \boldsymbol{\epsilon}_{intro} + \boldsymbol{\epsilon}_{new}, \tag{53}$$

$$\boldsymbol{\epsilon}_{intro} = \sqrt{\bar{\alpha}_{t-1}} \sigma_{\mathbf{y}} \mathbf{V} \Lambda_t \mathbf{V}^\top \mathbf{A}^\dagger \boldsymbol{\epsilon}_{\mathbf{n}}, \tag{54}$$

$$\boldsymbol{\epsilon}_{intro} + \boldsymbol{\epsilon}_{new} \sim \mathcal{N}(\mathbf{0}, \sigma_t^2 \mathbf{I}). \tag{55}$$

$\boldsymbol{\epsilon}_{intro}$ denotes the introduced noise, which can be further written as

$$\boldsymbol{\epsilon}_{intro} = \sqrt{\bar{\alpha}_{t-1}} \sigma_{\mathbf{y}} \mathbf{V} \Lambda_t \mathbf{V}^\top \mathbf{A}^\dagger \boldsymbol{\epsilon}_{\mathbf{n}} \tag{56}$$

$$\sim \mathcal{N}(\mathbf{0}, \bar{\alpha}_{t-1} \sigma_{\mathbf{y}}^2 (\mathbf{V} \Lambda_t \mathbf{V}^\top \mathbf{A}^\dagger) \mathbf{I} (\mathbf{V} \Lambda_t \mathbf{V}^\top \mathbf{A}^\dagger)^\top) \tag{57}$$

$$\sim \mathcal{N}(\mathbf{0}, \bar{\alpha}_{t-1} \sigma_{\mathbf{y}}^2 \mathbf{V} \Lambda_t \mathbf{V}^\top \mathbf{A}^\dagger (\mathbf{A}^\dagger)^\top \mathbf{V} \Lambda_t \mathbf{V}^\top) \tag{58}$$

$$\sim \mathcal{N}(\mathbf{0}, \bar{\alpha}_{t-1} \sigma_{\mathbf{y}}^2 \mathbf{V} \Lambda_t \mathbf{V}^\top \mathbf{V} \Sigma^\dagger \mathbf{U}^\top \mathbf{U} (\Sigma^\dagger)^\top \mathbf{V}^\top \mathbf{V} \Lambda_t \mathbf{V}^\top) \tag{59}$$

$$\sim \mathcal{N}(\mathbf{0}, \bar{\alpha}_{t-1} \sigma_{\mathbf{y}}^2 \mathbf{V} \Lambda_t \Sigma^\dagger (\Sigma^\dagger)^\top \Lambda_t \mathbf{V}^\top) \tag{60}$$

The variance matrix of $\boldsymbol{\epsilon}_{intro}$ can be simplified as $\mathbf{V} \mathbf{D}_t \mathbf{V}^\top$, with $\mathbf{D}_t = diag\{d_{t1}, d_{t2}, \cdots, d_{tD}\}$:

$$\boldsymbol{\epsilon}_{intro} \sim \mathcal{N}(\mathbf{0}, \mathbf{V} \mathbf{D}_t \mathbf{V}^\top), \quad d_{ti} = \begin{cases} \frac{\bar{\alpha}_{t-1} \sigma_{\mathbf{y}}^2 \lambda_{ti}^2}{s_i^2}, & s_i \neq 0, \\ 0, & s_i = 0 \end{cases}, \tag{61}$$

To construct $\boldsymbol{\epsilon}_{new}$, we define a new diagonal matrix $\boldsymbol{\Gamma}_t (= diag\{\gamma_{t1}, \gamma_{t2}, \cdots, \gamma_{tD}\})$:

$$\boldsymbol{\Gamma}_t = \sigma_t^2 \mathbf{I} - \mathbf{D}_t, \quad \gamma_{ti} = \begin{cases} \sigma_t^2 - \frac{\bar{\alpha}_{t-1} \sigma_{\mathbf{y}}^2 \lambda_{ti}^2}{s_i^2}, & s_i \neq 0, \\ \sigma_t^2, & s_i = 0 \end{cases}, \tag{62}$$

Now we can construct $\boldsymbol{\epsilon}_{new}$ by sampling from $\mathcal{N}(\mathbf{0}, \mathbf{V} \boldsymbol{\Gamma}_t \mathbf{V}^\top)$ to ensure that $\boldsymbol{\epsilon}_{intro} + \boldsymbol{\epsilon}_{new} \sim \mathcal{N}(\mathbf{0}, \mathbf{V}(\mathbf{D}_t + \boldsymbol{\Gamma}_t) \mathbf{V}^\top) = \mathcal{N}(\mathbf{0}, \sigma_t^2 \mathbf{I})$. An easier implementation is firstly sampling $\boldsymbol{\epsilon}_{temp}$ from $\mathcal{N}(\mathbf{0}, \boldsymbol{\Gamma}_t)$ and finally get $\boldsymbol{\epsilon}_{new} = \mathbf{V} \boldsymbol{\epsilon}_{temp}$. From Eq. 62, we also observe that $\lambda_{ti}$ guarantees the noise level of the introduced noise do not exceed the pre-defined noise level $\sigma_t$ so that we can get the formula of $\lambda_{ti}$ in $\Sigma_t (= \mathbf{V} \Lambda_t \mathbf{V}^\top, \Lambda_t = diag\{\lambda_{t1}, \lambda_{t2}, \cdots, \lambda_{tD}\})$:

$$\lambda_{ti} = \begin{cases} 1, & \sigma_t \geq \frac{\sqrt{\bar{\alpha}_{t-1}} \sigma_{\mathbf{y}}}{s_i}, \\ \frac{s_i \sigma_t \sqrt{1-\eta^2}}{\sqrt{\bar{\alpha}_{t-1}} \sigma_{\mathbf{y}}}, & \sigma_t < \frac{\sqrt{\bar{\alpha}_{t-1}} \sigma_{\mathbf{y}}}{s_i}, \\ 1, & s_i = 0, \end{cases}, \tag{63}$$

In the actual implementation, we have adopted the following formula for $\boldsymbol{\epsilon}_{temp}$ and it can be proved that its distribution is $\mathcal{N}(\mathbf{0}, \boldsymbol{\Gamma}_t)$:

$$\boldsymbol{\epsilon}_{temp}^{(i)} = \begin{cases} \sqrt{\sigma_t^2 - \frac{\bar{\alpha}_{t-1} \sigma_{\mathbf{y}}^2}{s_i^2}} \boldsymbol{\epsilon}^{(i)}, & \sigma_t \geq \frac{\sqrt{\bar{\alpha}_{t-1}} \sigma_{\mathbf{y}}}{s_i}, \\ \sigma_t \eta \boldsymbol{\epsilon}^{(i)}, & \sigma_t < \frac{\sqrt{\bar{\alpha}_{t-1}} \sigma_{\mathbf{y}}}{s_i}, \\ \sigma_t \sqrt{1-\eta^2} \mathcal{Z}_{\boldsymbol{\theta}}^{(i)} + \sigma_t \eta \boldsymbol{\epsilon}^{(i)}, & s_i = 0, \end{cases}, \tag{64}$$

where $\boldsymbol{\epsilon}_{temp}^{(i)}$ denotes the $i$-th element of the vector $\boldsymbol{\epsilon}_{temp}$ and $\boldsymbol{\epsilon} \sim \mathcal{N}(\mathbf{0}, \mathbf{I})$.

Note that the blue $\eta$ is not necessarily needed. By our theory in Sec. 3.3, $\eta$ should be 0 to maximize the preservation of range-space correction. But inspired by DDRM(Kawar et al., 2022), we find that involving $\eta$ help improves the robustness, though sacrificing some range-space information.

## J    ADDITIONAL RESULTS

We present additional quantitative results in Tab. 5, with corresponding visual results of DDNM in Fig. 11 and Fig. 12. Additional visual results of DDNM$^+$ are shown in Fig. 13 and Fig. 14. Additional results for real-world photo restoration are presented in Fig. 15. Note that all the additional results presented here do not use the time-travel trick.

| **CelebA-HQ** | 4× bicubic SR | | | | 8× bicubic SR | | | | 16× bicubic SR | | | |
|---|---|---|---|---|---|---|---|---|---|---|---|---|
| Method | PSNR↑ | SSIM↑ | *Cons*↓ | FID↓ | PSNR↑ | SSIM↑ | *Cons*↓ | FID↓ | PSNR↑ | SSIM↑ | *Cons*↓ | FID↓ |
| DDRM | 31.63 | 0.9452 | 33.88 | 31.04 | 28.11 | 0.9039 | 3.23 | 38.84 | 24.80 | 0.8612 | 0.36 | 46.67 |
| DDNM | 31.63 | 0.9450 | 4.80 | 22.27 | 28.18 | 0.9043 | 0.68 | 37.50 | 24.96 | 0.8634 | 0.10 | 45.5 |
| **ImageNet** | 4× bicubic SR | | | | 8× bicubic SR | | | | 16× bicubic SR | | | |
| Method | PSNR↑ | SSIM↑ | *Cons*↓ | FID↓ | PSNR↑ | SSIM↑ | *Cons*↓ | FID↓ | PSNR↑ | SSIM↑ | *Cons*↓ | FID↓ |
| DDRM | 27.38 | 0.8698 | 19.79 | 43.15 | 23.75 | 0.7668 | 2.70 | 83.67 | 20.85 | 0.6842 | 0.38 | 130.81 |
| DDNM | 27.46 | 0.8707 | 4.92 | 39.26 | 23.79 | 0.7684 | 0.72 | 80.15 | 20.90 | 0.6853 | 0.11 | 128.13 |
| **CelebA-HQ** | inpainting (Mask 1) | | | | inpainting (Mask 2) | | | | inpainting (Mask 3) | | | |
| Method | PSNR↑ | SSIM↑ | *Cons*↓ | FID↓ | PSNR↑ | SSIM↑ | *Cons*↓ | FID↓ | PSNR↑ | SSIM↑ | *Cons*↓ | FID↓ |
| DDRM | 34.79 | 0.9783 | 1325.46 | 12.53 | 38.27 | 0.9879 | 1357.09 | 10.34 | 35.77 | 0.9767 | - | 21.49 |
| DDNM | 35.64 | 0.9823 | 0.0 | 4.54 | 39.38 | 0.9915 | 0.0 | 2.82 | 36.32 | 0.9797 | - | 12.46 |
| **ImageNet** | inpainting (Mask 1) | | | | inpainting (Mask 2) | | | | inpainting (Mask 3) | | | |
| Method | PSNR↑ | SSIM↑ | *Cons*↓ | FID↓ | PSNR↑ | SSIM↑ | *Cons*↓ | FID↓ | PSNR↑ | SSIM↑ | *Cons*↓ | FID↓ |
| DDRM | 31.73 | 0.9663 | 876.86 | 4.82 | 34.60 | 0.9785 | 1036.85 | 3.77 | 31.34 | 0.9439 | - | 12.84 |
| DDNM | 32.06 | 0.9682 | 0.0 | 3.89 | 34.92 | 0.9801 | 0.0 | 3.19 | 31.62 | 0.9461 | - | 9.73 |
| **CelebA-HQ** | deblur (Gaussian) | | | | deblur (anisotropic) | | | | deblur (uniform) | | | |
| Method | PSNR↑ | SSIM↑ | *Cons*↓ | FID↓ | PSNR↑ | SSIM↑ | *Cons*↓ | FID↓ | PSNR↑ | SSIM↑ | *Cons*↓ | FID↓ |
| DDRM | 43.07 | 0.9937 | 297.15 | 6.24 | 41.29 | 0.9909 | 312.14 | 7.02 | 40.95 | 0.9900 | 182.27 | 7.74 |
| DDNM | 46.72 | 0.9966 | 60.00 | 1.41 | 43.19 | 0.9931 | 66.14 | 2.80 | 42.85 | 0.9923 | 41.86 | 3.79 |
| **ImageNet** | deblur (Gaussian) | | | | deblur (anisotropic) | | | | deblur (uniform) | | | |
| Method | PSNR↑ | SSIM↑ | *Cons*↓ | FID↓ | PSNR↑ | SSIM↑ | *Cons*↓ | FID↓ | PSNR↑ | SSIM↑ | *Cons*↓ | FID↓ |
| DDRM | 43.01 | 0.9921 | 207.90 | 1.48 | 40.01 | 0.9855 | 221.23 | 2.55 | 39.72 | 0.9829 | 134.60 | 3.73 |
| DDNM | 44.93 | 0.9937 | 59.09 | 1.15 | 40.81 | 0.9864 | 63.89 | 2.14 | 40.70 | 0.9844 | 41.86 | 3.22 |
| **CelebA-HQ** | CS (ratio=0.5) | | | | CS (ratio=0.25) | | | | | | | |
| Method | PSNR↑ | SSIM↑ | *Cons*↓ | FID↓ | PSNR↑ | SSIM↑ | *Cons*↓ | FID↓ | | | | |
| DDRM | 31.52 | 0.9520 | 2171.76 | 25.71 | 24.86 | 0.8765 | 1869.03 | 46.77 | | | | |
| DDNM | 33.44 | 0.9604 | 1640.67 | 15.81 | 27.56 | 0.9090 | 1511.51 | 28.80 | | | | |
| **ImageNet** | CS (ratio=0.5) | | | | CS (ratio=0.25) | | | | | | | |
| Method | PSNR↑ | SSIM↑ | *Cons*↓ | FID↓ | PSNR↑ | SSIM↑ | *Cons*↓ | FID↓ | | | | |
| DDRM | 26.94 | 0.8902 | 6293.69 | 25.01 | 19.95 | 0.7048 | 3444.50 | 97.99 | | | | |
| DDNM | 29.22 | 0.9106 | 5564.00 | 18.55 | 21.66 | 0.7493 | 3162.30 | 64.68 | | | | |
| **CelebA-HQ** | Colorization | | | | | | | | | | | |
| Method | PSNR↑ | SSIM↑ | *Cons*↓ | FID↓ | | | | | | | | |
| DDRM | 26.38 | 0.7974 | 455.90 | 31.26 | | | | | | | | |
| DDNM | 26.25 | 0.7947 | 48.87 | 26.44 | | | | | | | | |
| **ImageNet** | Colorization | | | | | | | | | | | |
| Method | PSNR↑ | SSIM↑ | *Cons*↓ | FID↓ | | | | | | | | |
| DDRM | 23.34 | 0.6429 | 260.43 | 36.56 | | | | | | | | |
| DDNM | 23.47 | 0.6550 | 42.32 | 36.32 | | | | | | | | |

Table 5: Comprehensive quantitative comparisons between DDNM and DDRM.

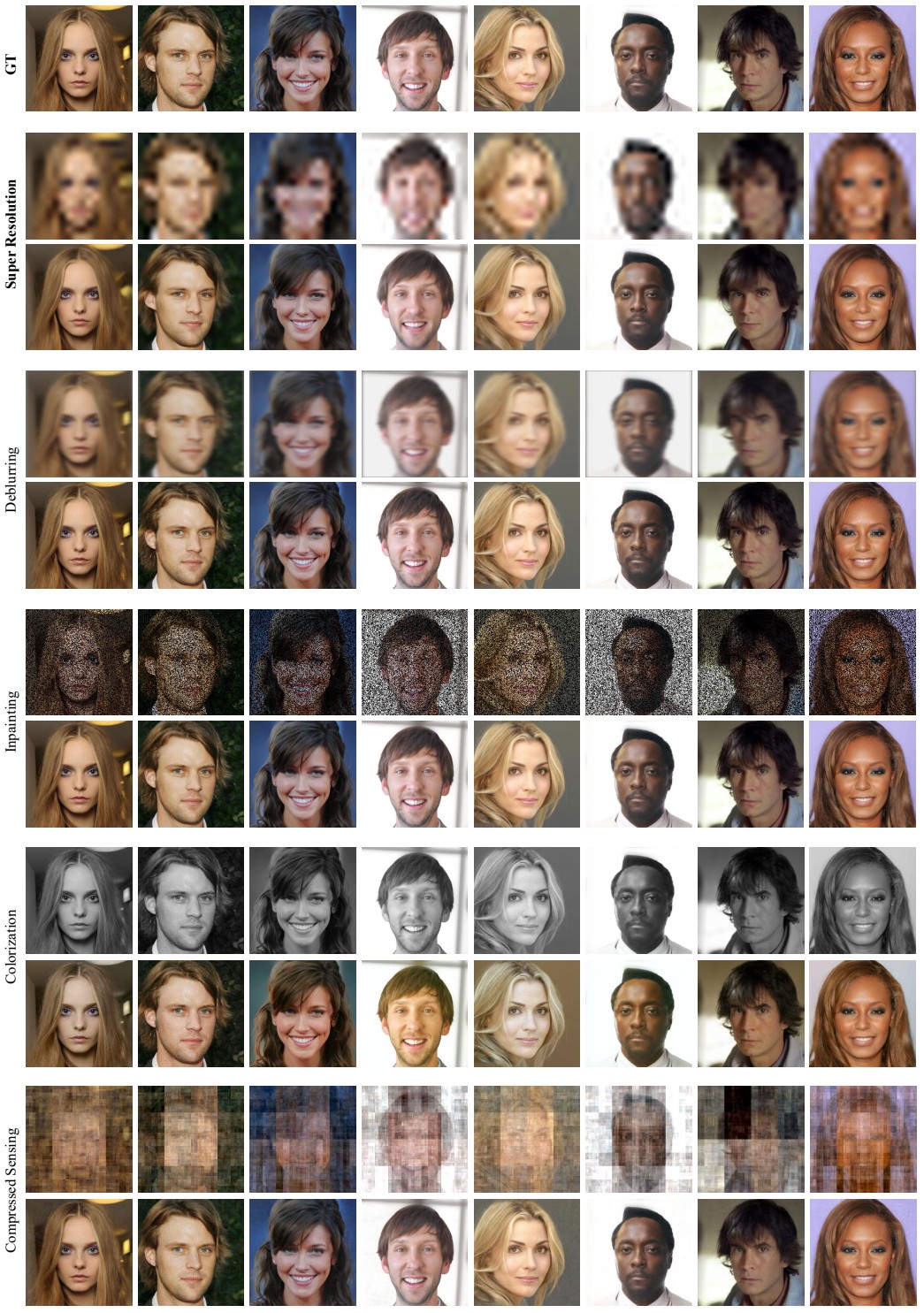

Figure 11: Image restoration results of DDNM on CelebA-HQ.

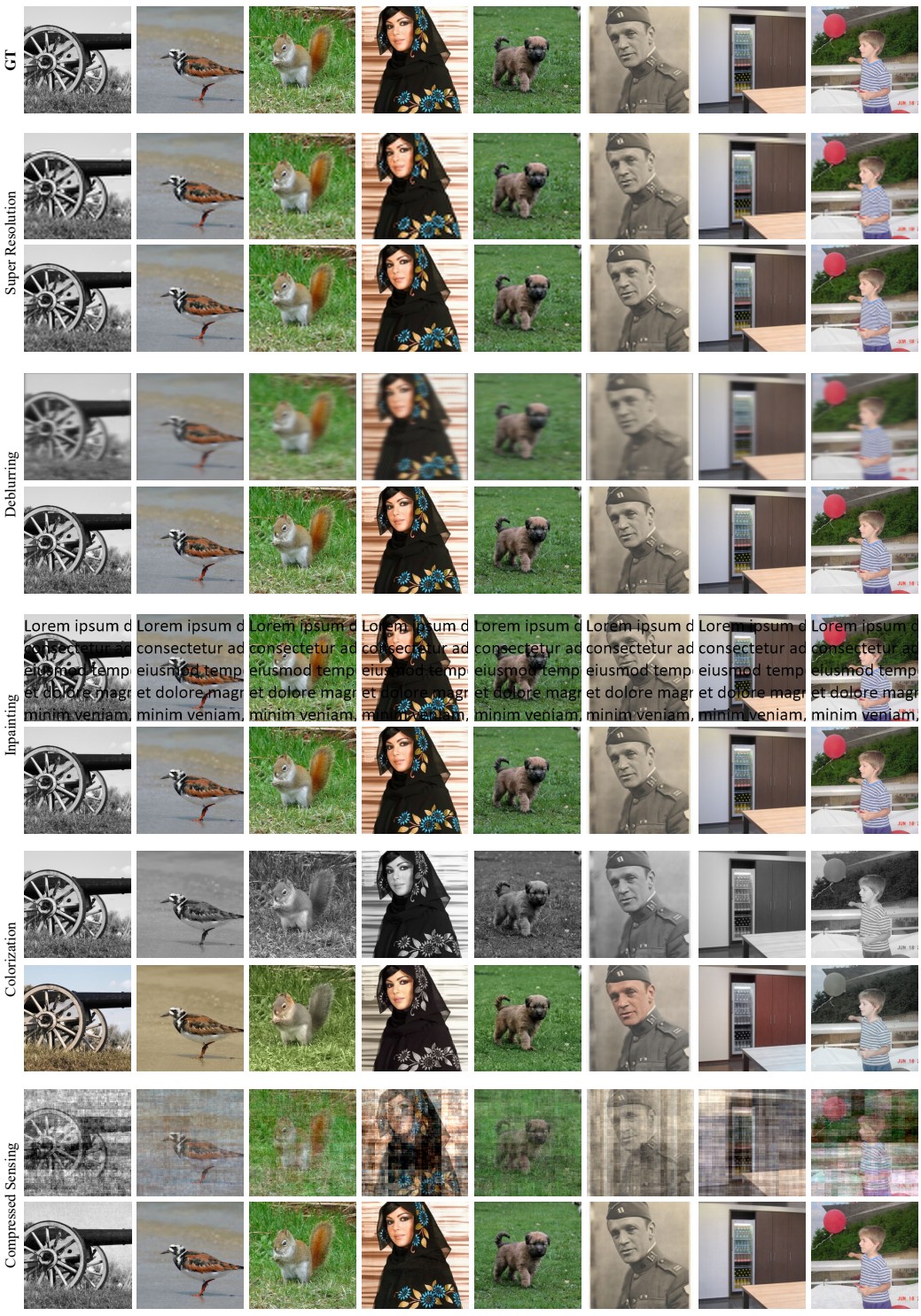

Figure 12: Image restoration results of DDNM on ImageNet.

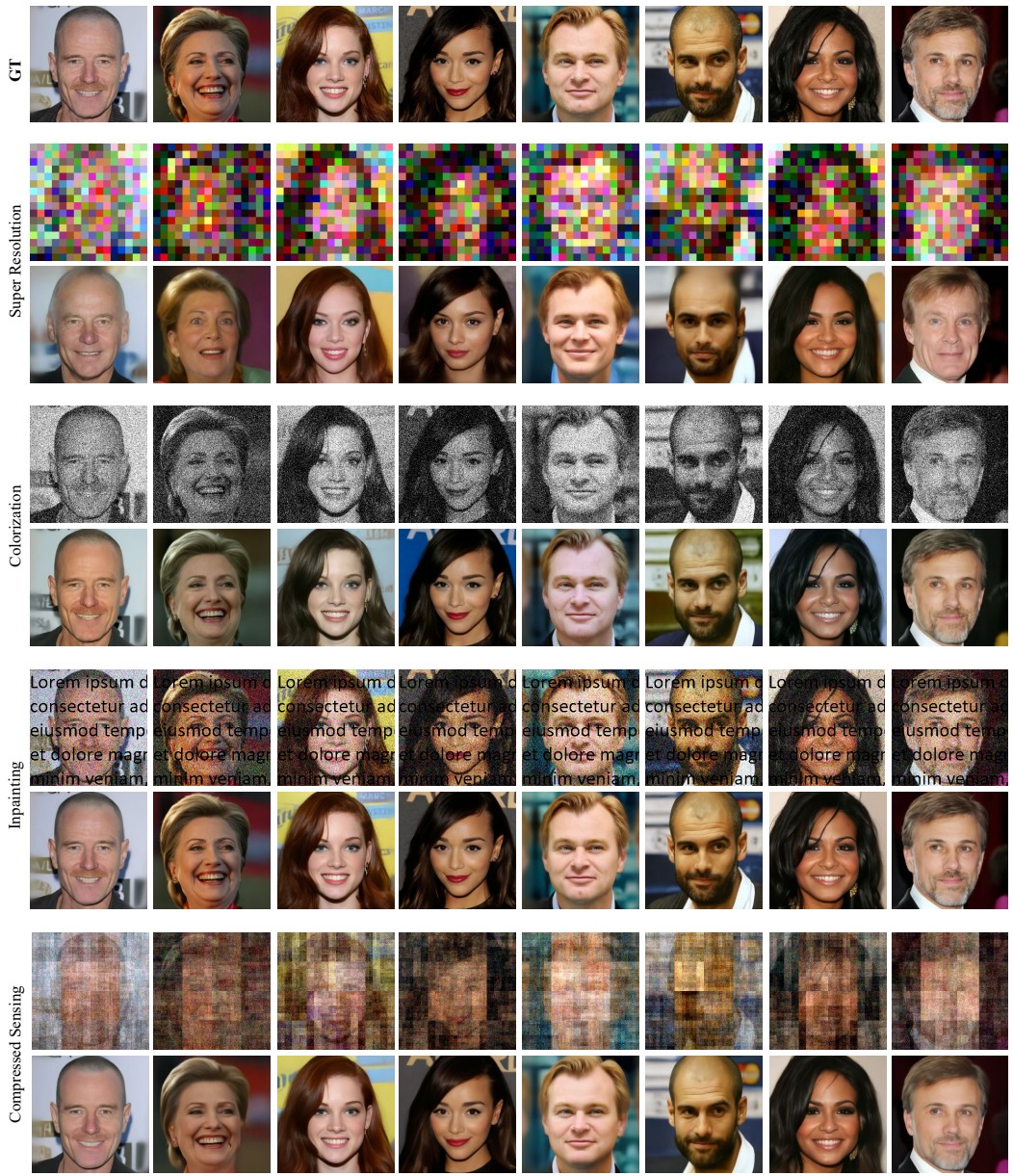

Figure 13: Noisy image restoration results of DDNM$^+$ on CelebA-HQ. The results here do not use the time-travel trick.

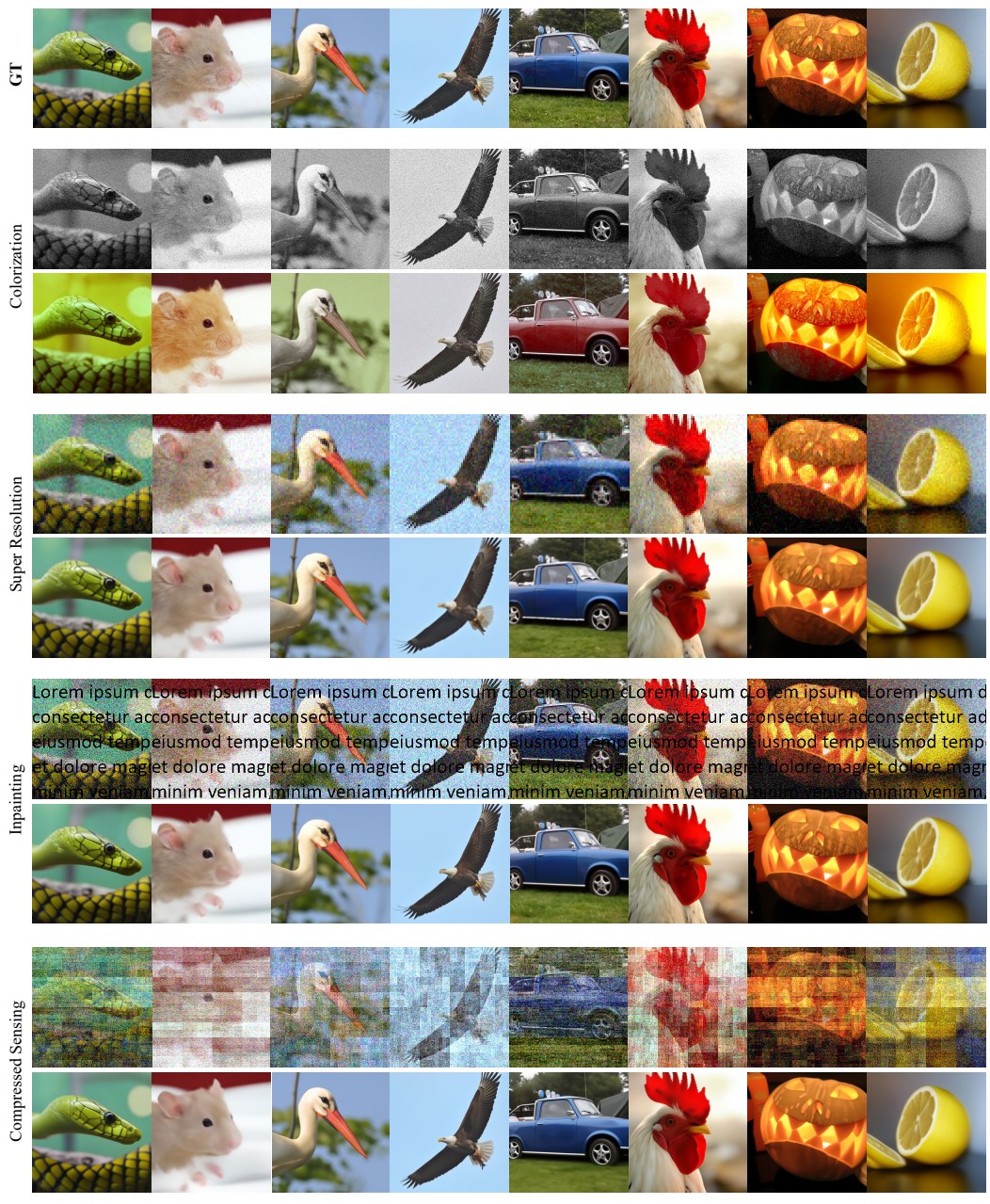

Figure 14: Noisy image restoration results of DDNM$^+$ on ImageNet. The results here do not use the time-travel trick.

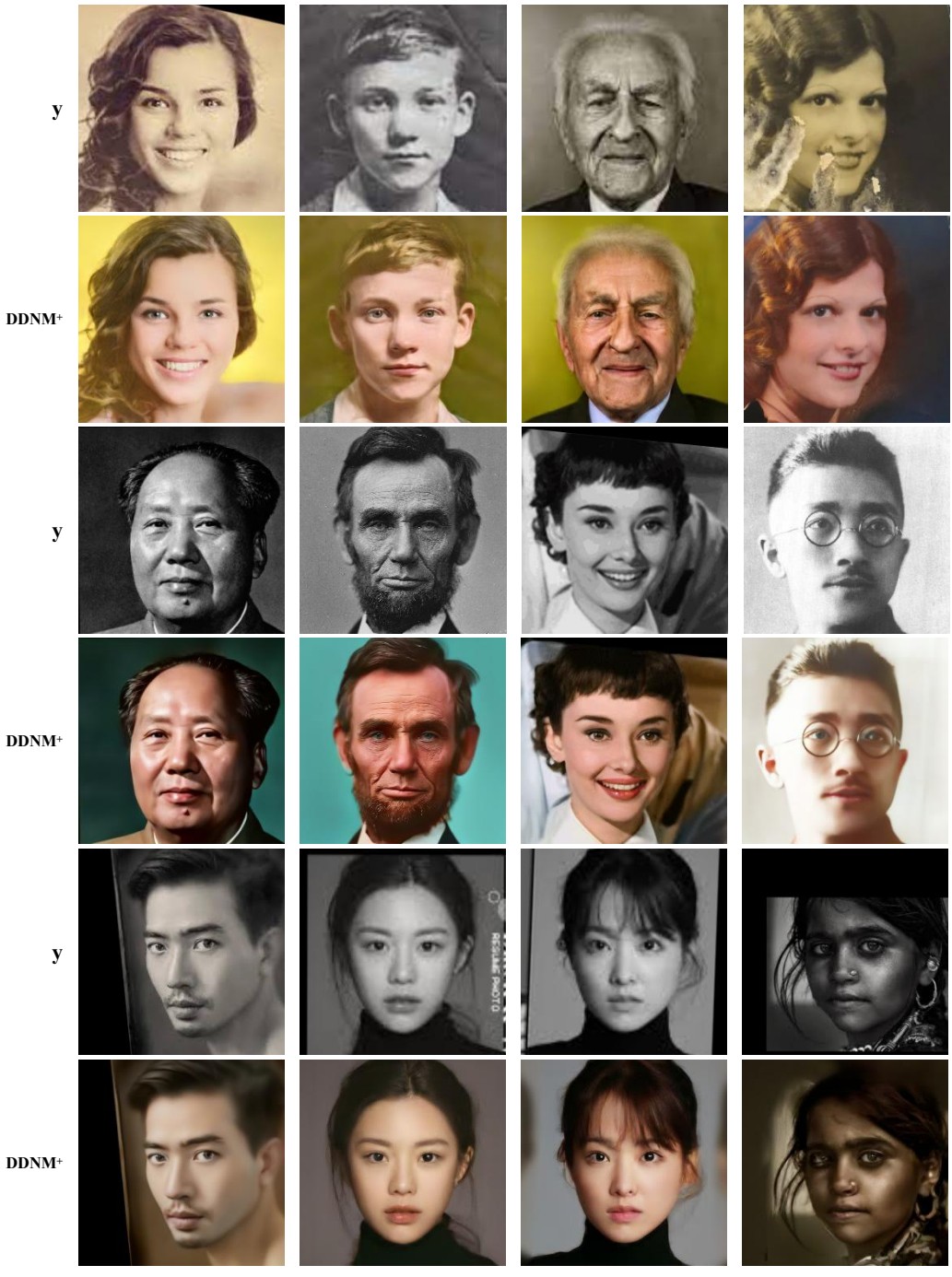

Figure 15: Restoring real-world photos using DDNM. **y** represents the degraded images collected from the internet. The results here do not use the time-travel trick.

