# OpenReview forum: "Zero-Shot Image Restoration Using Denoising Diffusion Null-Space Model"
_ICLR.cc/2023/Conference — ICLR 2023 notable top 25%_

### Official Review · Reviewer_hK3i · 2022-10-17

**Confidence:** 3
**Correctness:** 3
**Technical Novelty And Significance:** 2
**Empirical Novelty And Significance:** 2
**Recommendation:** 8

**Clarity, Quality, Novelty And Reproducibility:**

In method section, authors provided clear description about their claims on Null space refining and summarized well in pseudo codes at Algorithm1 & 2. Except ablation study on effectiveness of Null space projection lacks. Comparisons with DDRM may be different from such ablation studies.

The quality of results is fine when we review result table 1 and supplementaries. However, I have some doubts on comparison with other supervised learning methods. Basically, their model, DDNM, is also necessary to see ground-truth set for training a generative model, so it would be similar situation with supervised learning setup: train on ground-truth set and then inference on unseen testset. Thus, it may be important to see comparisons with supervised learning baselines in result tables (currently, this paper is compared only with DDPM-based generative models.

In the perspective of novelty, this paper reminds me following paper,
Song, Yang, et al. "Solving inverse problems in medical imaging with score-based generative models." ICLR (2022).
This paper also introduced SDE for inverse problems and they use very much similar steps for sampling. Major difference from this work is the existence of null space projection of target x in reverse process. I'm convinced of the superiority of this manuscript upon the mentioned paper.

The paper contains detailed pseudo codes enough to be implemented, however, there is no actual source code and information for neural network's architectures.

**Strength And Weaknesses:**

They introduced null-space decomposition during reverse process in DDPM which helps to pursue robust data consistency and realistic image reconstruction. The detailed descriptions toward it are given in Algorithmic tables and associated illustration. It's easy to understand. Their supported results in Figure 3 and 4 are quite good qualities. The results seem to be beyond the bar of quality in ICLR.

However, I'm a bit afraid that the result seems to be far from original contents, which brings fake information such as figure 3 at 32x super-resolution on CelebA. It looks like a posterior sample not like a reconstruction results. In my opinion, 32x SR results can not be called as reconstruction, it's a synthetic image. DDNM provides 32x32 pixels out of 1pixel, and it's hard to avoid referring contents from synthetic textures of training manifold. Therefore, especially for face datasets, I couldn't accept it as recovered result. It is closer to synthesizing images irrelevant with ground-truth. Thus, I recommend the authors to provide residual images and ground-truth images  in super resolution example and other examples, too.

**Summary Of The Paper:**

This paper proposed denoising diffusion probabilistic model for image restoration. They specially applied null-space projection at every iteration to target variable x and used A^T y as range spaced values. By considering only null-space contents in reverse process, they achieved realistic images with data consistency. They demonstrated several image restoration applications: super-resolution, inpainting, colorization

**Summary Of The Review:**

In general, this paper is well written and supported by remarkable results. The idea using null space component for residual components in reverse process in DDPM seems to be interesting. However, their approach is similar with the paper from ICLR 2022(above). Unless I see the performance difference with the ICLR 2022 paper or proper explanation of distinguishable points, I would be a doubt on effectiveness of null space projection.

Also, I have a concern that reconstructions from severe conditions go beyond image restoration and become closer to arbitrary image synthesizing. To avoid such concerns, the results from supervised learning need to be demonstrated together.

---

> ### Author Response · Authors · 2022-11-14
> **Response to Reviewer hK3i (Part 1)**
>
> Thanks for your constructive discussion and suggestions!
>
> ### **Q: "Similar to Song's method"**
>
> A: Thanks for the question! The key step of Song's method\[3]  is: $\hat{\mathbf{x}}\_{t}=\mathbf{T^{-1}}[\lambda\mathbf{\Lambda}\mathbf{P^{-1}}(\mathbf{\Lambda})\mathbf{y}\_{t}+(1-\lambda)\mathbf{\Lambda}\mathbf{T}\mathbf{x}\_{t}+(\mathbf{I}-\mathbf{\Lambda})\mathbf{T}\mathbf{x}\_{t}]$, where the degradation operator $\mathbf{A}=\mathbf{P}(\mathbf{\Lambda})\mathbf{T}$, corresponds to CT/MRI applications.
>
> We are sure that **our work is superior to Song's method [3] in a lot of ways**:
>
> 1. The first point is that their formulation can be seen **as a special case of ours**.  When bring their equation into clean $\mathbf{x}\_{0|t}$ domain and considering noise-free situation where $\lambda=1$, it can be simplified as our formulation: $\hat{\mathbf{x}}\_{0|t}=\mathbf{A^{\dagger}}\mathbf{y} + (\mathbf{I} - \mathbf{A^{\dagger}}\mathbf{A})\mathbf{x}\_{0|t}$, where $\mathbf{A}^{\dagger}=\mathbf{T^{-1}}\mathbf{P^{-1}}(\mathbf{\Lambda})$. It is worth noting that our formulation is more generalized since we do not limit the form of $\mathbf{A}$ and $\mathbf{A}^{\dagger}$, as long as they are linear and satisfy $\mathbf{A}\mathbf{A}^{\dagger}\mathbf{A}=\mathbf{A}$. In our method, $\mathbf{A}$ and $\mathbf{A}^{\dagger}$ can be hand-designed (as we presented in Section 3.2), SVD-based, Fourier-based, or even a learned one. Besides, our motivation and derivation are more concise and insightful.
> 2. The second point is that **we operate on clean $\mathbf{x}\_{0|t}$ domain** while they operate on noisy $\mathbf{x}\_{t}=a_t\mathbf{x}\_{0|t}+b_t\boldsymbol{\epsilon}\_1$ domain, thus they need noised $\mathbf{y}\_{t}=a_t\mathbf{y}+b_t\boldsymbol{\epsilon}\_2$ but we don't. Note that $\boldsymbol{\epsilon}\_1,\boldsymbol{\epsilon}\_2\sim\mathcal{N}(\mathbf{0}, \mathbf{I})$, $a_t$ and $b_t$ denote the scale factor). For noise-free situation, we can rewrite Song's method as $\hat{\mathbf{x}}\_{t}=\mathbf{A^{\dagger}}\mathbf{y}\_t + (\mathbf{I} - \mathbf{A^{\dagger}}\mathbf{A})\mathbf{x}\_{t}=\mathbf{A^{\dagger}}(a_t\mathbf{y}+b_t\boldsymbol{\epsilon}\_2) + (\mathbf{I} - \mathbf{A^{\dagger}}\mathbf{A})(a_t\mathbf{x}\_{0|t}+b_t\boldsymbol{\epsilon}\_1)=a_t\hat{\mathbf{x}}\_{0|t} + b_t\boldsymbol{\epsilon}\_1 + b_t\mathbf{A^{\dagger}}(\boldsymbol{\epsilon}\_2-\mathbf{A}\boldsymbol{\epsilon}\_1)$, where $\hat{\mathbf{x}}\_{0|t}=\mathbf{A^{\dagger}}\mathbf{y} + (\mathbf{I} - \mathbf{A^{\dagger}}\mathbf{A})\mathbf{x}\_{0|t}$. As is proved in our paper, the term $\hat{\mathbf{x}}\_{0|t}$ conforms to data consistency since $\mathbf{A}\hat{\mathbf{x}}\_{0|t}=\mathbf{y}$, hence $a_t\hat{\mathbf{x}}\_{0|t} + b_t\boldsymbol{\epsilon}\_1$ is the correct distribution of $\hat{\mathbf{x}}\_{t}$ while $b_t\mathbf{A^{\dagger}}(\boldsymbol{\epsilon}\_2-\mathbf{A}\boldsymbol{\epsilon}\_1)$ is the involved extra noise, which does not equal to 0 when $t\neq 0$, thus harms the restoration quality during the generation. The experiments below support this well.
>
>     \*experiment setting： The results are directly copied from Table 1 in the paper, since ILVR[10] and RePaint[11] are special cases of DDNM on SR and Inpainting tasks, respectively. The only difference is that they operate on noisy $\mathbf{x}\_{t}$ domain, as done in Song's method.
>
>     **4xSR**
>     | Baseline |  Domain| PSNR↑/FID↓  |
>     |  ----  |  ----  | ----  |
>     |  ILVR[10]  | $\mathbf{x}\_{t}$ (Song's) | 27.40/43.66 |
>     |  DDNM  | $\mathbf{x}\_{0\|t}$ (ours) | **27.46/39.26** |
>
>     **Inpainting**
>     | Baseline |  Domain| PSNR↑/FID↓  |
>     |  ----  |  ----  | ----  |
>     |  RePaint[11]  | $\mathbf{x}\_{t}$ (Song's) | 31.87/12.31 |
>     |  DDNM  | $\mathbf{x}\_{0\|t}$ (ours) | **32.06/3.89** |

---

> > ### Author Response · Authors · 2022-11-14
> > **Response to Reviewer hK3i (Part 2)**
> >
> > 3. The third point is that **we provide a better solution for noisy tasks**. They use a scalar $\lambda$ to scale down the range-space correction, so as to scale down the noise. We have proved in Section 3.3 that this still involves additional noise in the generation process. Consider $\mathbf{y}=\mathbf{A}\mathbf{x}+\mathbf{n}$, where $\mathbf{n}\sim\mathcal{N}(\mathbf{0}, \sigma_{\mathbf{y}}\mathbf{I})$, we can rewrite Song's formulation as $\hat{\mathbf{x}}\_{t}=\mathbf{x}\_{t} + \lambda\mathbf{A^{\dagger}}(\mathbf{y} - \mathbf{A}\mathbf{x}\_{t})=\mathbf{x}\_{t} + \lambda\mathbf{A^{\dagger}}(\mathbf{A}\mathbf{x} - \mathbf{A}\mathbf{x}\_{t}) + \lambda\mathbf{A^{\dagger}}\mathbf{n}$, where $\lambda\mathbf{A}^{\dagger}\mathbf{n}$ is the involved extra noise, which will **harm the generation quality**. Besides, Song's paper does not show performance on noisy tasks. Instead, we use $\Sigma_t$ to scale down the range-space correction, and use $\Phi_t$ to accurately scale the added noise in $\mathbf{x}\_{t}$ to make sure the total noise is invariant, thus making restoration quality comparable to that of noise-free situations. $\Sigma_t$ and $\Phi_t$ can be accurately calculated according to the noise level $\sigma$ and time-step $t$. (see Section 3.3 and Appendix I for details).
> >
> >     \*experiment setting：DDNM uses the same setting as Q1 in the overall response unless otherwise specified. We find Song's method yields very bad results when facing high noise like $\sigma_{\mathbf{y}}$=0.2 no matter what $\lambda$ we set. Hence we set $\lambda_t$ to decay with time, which yields much better results but is still inferior to ours.
> >
> >     **Compressed Sensing, ratio=0.25, $\sigma\_{\mathbf{y}}$=0.2**
> >     | Baseline | Domain | Denoising | PSNR↑/FID↓  |
> >     |  ----  |  ----  |  ----  | ----  |
> >     |  DDNM  | $\mathbf{x}\_{0\|t}$ (ours) | $\lambda$ (Song's) | 18.23/110.38 |
> >     |  DDNM  | $\mathbf{x}\_{0\|t}$ (ours) | $\Sigma_t, \Phi_t$ (ours) | **20.69/88.79** |
> >
> >     **4xSR, $\sigma\_{\mathbf{y}}$=0.2**
> >     | Baseline | Domain |  Denoising  | PSNR↑/FID↓  |
> >     |  ----  |  ----  |  ----  | ----  |
> >     |  DDNM  | $\mathbf{x}\_{0\|t}$ (ours) |  $\lambda$ (Song's) | 20.23/111.55 |
> >     |  DDNM  | $\mathbf{x}\_{0\|t}$ (ours) | $\Sigma_t, \Phi_t$ (ours) | **22.67/80.69** |
> >
> > 4. The fourth point is that we propose a **time-travel trick** that can improve restoration quality for all mentioned IR tasks. Here we paste the ablation results from our paper.
> >
> >     **Compressed Sensing, ratio=0.1**
> >     | Baseline |  Time-Travel | PSNR↑/FID↓  |
> >     |  ----  |  ----  | ----  |
> >     |  DDNM  |  NO (Song's) | 15.74/110.7 |
> >     |  DDNM  | YES (ours) | **26.33/47.93** |
> >
> > 5. Besides, Song's work only experiments on CT/MRI, but we demonstrate **variety of applications** and provide a concise and robust solution for real-world image restoration.
> >
> >
> > To conclude, **whether in theory or performance, we have significant advantages over Song's method**.

---

> > > ### Comment · Reviewer_hK3i · 2022-11-26
> > > **response to rebuttal letter**
> > >
> > > Thanks to author for addressing my previous concerns.
> > > The concerns were properly treated, so I raised my scores.

---

> ### Author Response · Authors · 2022-11-14
> **Response to Reviewer hK3i (Part 3)**
>
> ### **Q: "32x super-resolution result seems to be far from original contents"**
>
> A: Thanks for the question! A general linear degradation process (noise-free) can be formulated as $\mathbf{y}=\mathbf{A}\mathbf{x}$. Usually, the degradation operator $\mathbf{A}$ leads to dimensionality reduction, which means information loss. So a good image restoration (IR) result $\hat{\mathbf{x}}$ conforms to two properties:
> 1. $\textit{Consistency}: \quad \mathbf{A}\hat{\mathbf{x}} \equiv \mathbf{y}$
> 2. $\textit{Realness}: \quad \hat{\mathbf{x}} \sim q(\mathbf{x})$
>
> We are sure this describes a common goal for linear IR tasks. The $\textit{Consistency}$ is the only bridge between the ground truth $\mathbf{x}$ and the restored results $\hat{\mathbf{x}}$, i.e., $\mathbf{A}\hat{\mathbf{x}} \equiv \mathbf{y}\equiv \mathbf{A}\mathbf{x}$. Note that DDNM analytically assures the $\textit{Consistency}$ through range-null space decomposition. No matter what downsampling operator $\mathbf{A}$ is used, 32x downsampling will lead to 1024 times dimensionality reduction. In this case, there is no way to guarantee that the result $\hat{\mathbf{x}}$ will be similar to the original image. In fact, there are countless results that satisfy $\textit{Consistency}$ and $\textit{Realness}$ constraint, they are all good results.
>
> ### **Q: "Ablation study on effectiveness of Null space projection"**
>
> A: Thanks for the suggestion! The critical step in DDNM is $\hat{\mathbf{x}}\_{0|t}=\mathbf{A^{\dagger}}\mathbf{y} + (\mathbf{I} - \mathbf{A^{\dagger}}\mathbf{A})\mathbf{x}\_{0|t}$, where $(\mathbf{I} - \mathbf{A^{\dagger}}\mathbf{A})$ can be seen as the null-space projection, and is critical for assuring the $\textit{Consistency}$ since $\mathbf{A}\hat{\mathbf{x}}\_{0|t}=\mathbf{A}\mathbf{A^{\dagger}}\mathbf{y} + \mathbf{A}(\mathbf{I} - \mathbf{A^{\dagger}}\mathbf{A})\mathbf{x}\_{0|t}=\mathbf{y}$. If without the null-space projection, it becomes $\hat{\mathbf{x}}\_{0|t}=\mathbf{A^{\dagger}}\mathbf{y} + \mathbf{x}\_{0|t}$, which yields random results that does not conforms to $\textit{Consistency}$ at all.
>
> ### **Q: "Comparing to supervised methods"**
>
> A: Thanks for the suggestion! We have presented the analysis and experiment results in our overall response.
>
> ### **Q: "No actual source code and information for neural network's architectures."**
>
> A: Thanks for the question! Our method is independent of the network architecture, hence we do not provide network details. But we do mention in Section 4.1 the pretrained diffusion model we used. Interested readers can find details of these models by reference. We have added Pytorch-like code to Appendix E, and are working on a corresponding GitHub project.
>
> ### **Q: "Inaccurate claim"**
>
> A: Thanks for the reminder! We have checked and revised the claims.

---

### Official Review · Reviewer_9BqY · 2022-10-21

**Confidence:** 4
**Correctness:** 4
**Technical Novelty And Significance:** 3
**Empirical Novelty And Significance:** 3
**Recommendation:** 8

**Clarity, Quality, Novelty And Reproducibility:**

The work is clearly explained and the idea is novel and sound. The explanation in section 3 seems to be self-contained enough to allow re-implementation of the method. The authors are highly encouraged to publicly release their original code.

**Strength And Weaknesses:**

Strengths:
- Interesting and novel idea to specialize the use of pretrained DDPMs for zero-shot solution of linear inverse problems
- Excellent experimental results on various tasks

Weaknesses:
- It would be interesting to add a comparison of the proposed method with supervised discriminative models (e.g., SwinIR) trained for a specific restoration task to understand how far behind generative models are
- The authors acknowledge that computational complexity is a limitation of DDNM. However, they should also provide some experimental metrics such as the method runtime compared to existing methods

**Summary Of The Paper:**

The authors propose a novel method for zero-shot solution of image restoration problem formulated as linear inverse problems by modifying denoising diffusion models. In particular, they propose to only update the null space of the forward operator during the reverse process of the DDPM. Overall, the idea is novel, clearly explained and a significant amount of good experimental results on various problems is presented to support the work.

**Summary Of The Review:**

The paper presents a novel method based on DDPMs for zero-shot image restoration. The technique is novel and shows very competitive performance against other techniques based on generative models. The work is very valuable as an alternative to supervised training of discriminative models but a point of comparison between them should be included.

---

> ### Author Response · Authors · 2022-11-14
> **Response to Reviewer 9BqY**
>
> Thanks for your constructive suggestions!
>
> ### **Q: "Comparing to supervised methods"**
>
> A: Thanks for your advice! We surprisingly find DDNM outperforms SwinIR by a large margin! You may refer to our overall response for experiment results.
>
> ### **Q: "Time efficiency & memory efficiency"**
>
> A: Thanks for your advice! We have concluded the updates and results in our overall response.

---

### Official Review · Reviewer_exTL · 2022-10-23

**Confidence:** 4
**Correctness:** 3
**Technical Novelty And Significance:** 3
**Empirical Novelty And Significance:** 3
**Recommendation:** 6

**Clarity, Quality, Novelty And Reproducibility:**

The clarity, quality and novelty of this paper is good. The reproducibility needs further improvement.

**Strength And Weaknesses:**

Strength:

(1)This paper has detailed background introduction and theoretical analysis about using diffusion model to solve diverse IR problems.

(2)This paper provides convincing proof and introduction about the usage of the generated null-space contents for image restoration.

(3)Another enhanced version DDNM+ is further proposed to handle noisy IR tasks. A time-travel trick is designed to provide better restoration quality.

(4)The provided quantitative and visual results validate the effectiveness of the proposed DDNM, which are superior over existing state-of-the-art zero-shot IR methods.

Weaknesses:

(1)The author mention the limitation of time efficiency in the appendix part. How about the memory efficiency?

(2)It is a little too absolute to claim that “solve arbitrary IR tasks with arbitrary degrees of degradation” in the introduction of contributions, since the degradation operators are asked to be linear.


**Summary Of The Paper:**

This paper proposes a novel zero-shot framework for solving diverse linear image restoration (IR) problems, named Denoising Diffusion Null-Space Model (DDNM). Firstly, it provides detailed theory analysis for using a pre-trained diffusion model to solve linear IR problems. Secondly, the proposed DDNM can well solve those IR tasks with arbitrary degrees of degradation and also support solving noisy IR tasks.

**Summary Of The Review:**

In my opinion, this paper is marginally above the acceptance threshold based on the strengths and weaknesses.

---

> ### Author Response · Authors · 2022-11-14
> **Response to Reviewer exTL**
>
> Thanks for your constructive suggestions!
>
> ### **Q: "Time efficiency & memory efficiency"**
>
> A: Thanks for your advice! We have concluded our updates and results in our overall response.
>
> ### **Q: "Inaccurate claim"**
>
> A: Thanks for the reminder! We revised the sentence "solve arbitrary IR tasks" as "solve arbitrary *linear* IR tasks". We also checked the related parts.

---

> ### Author Response · Authors · 2022-12-05
> **Sincerely look forward to your response**
>
> Dear reviewer,
>
> We have added comparisons of time & memory efficiency to Appendix A, and revised the overclaims you mentioned. Besides, we have released the source code and provided datasets and detailed instructions for researchers to reproduce the results in the paper.
>
> We would appreciate it if you could re-evaluate our work! Thanks.

---

> > ### Comment · Reviewer_exTL · 2022-12-05
> > **response to rebuttal**
> >
> > Thanks for the detailed feedback, which addresses my concerns. I keep my original rating score and recommend to accept it.

---

### Official Review · Reviewer_NtKw · 2022-10-25

**Confidence:** 3
**Correctness:** 3
**Technical Novelty And Significance:** 3
**Empirical Novelty And Significance:** 3
**Recommendation:** 8

**Clarity, Quality, Novelty And Reproducibility:**

The contribution of this work is good since it constructs a general restoration model for arbitrary linear degradations.

**Strength And Weaknesses:**

Strength:
1.	This work proposes a unified theoretical framework for linear image restoration problems. Specifically, the sample x is decomposed into the range-space and null-space of A, which is the linear degradation operation. Then, the null-space is refined iteratively.
2.	The proposed method can be utilized in many different tasks and the results are promising.
Weaknesses:

1.	As demonstrated by the authors, DDNM+ can solve any real-world IR task as long as we we can construct an approximate linear degradation A and its pseudo-inverse A^-1. However, for real image degradation, A is most likely non-linear. For example, realistic noise in sRGB domain is nonlinear. If we directly utilize this method to solve real noise removal problem, I guess the performance will be heavily degraded.

2.	The computing complexity. The authors should give the inference time to let readers aware of the computing complexity of the proposed method.


**Summary Of The Paper:**

This work proposed a novel zero-shot framework for linear image restoration, and the key module is denoising diffusion null-space model. It only requires a pretrained diffusion model, without further training and optimization. Experimental results show that the proposed method outperforms sota zero-shot restoration methods in many tasks.

**Summary Of The Review:**

Considering the novelty of this work and the good performance, I recommend accept.

After rebuttal, the authors have addressed my concerns. Therefore, I still recommend accept.

---

> ### Author Response · Authors · 2022-11-14
> **Response to Reviewer NtKw**
>
> Thanks for your constructive discussion and suggestions!
>
> ### **Q: "DDNM+ on real-world IR tasks"**
>
> A: That's an interesting topic! Actually, DDNM+ can well handle old photo restoration, where the degradation operator $\mathbf{A}$ is unknown and non-linear and even contains non-Gaussian noise. Our solution is based on the following observations:
> 1. DDNM+ is designed to eliminate the noise in range-space and is robust to diverse types of noise.
> 2. For $\mathbf{y}$ in the image domain, e.g., old photos or real-world degraded photos, the non-linear artifacts can generally be divided into **global** (e.g., the real-world noise in Fig1c ) and **local** (e.g., the scratches in Fig1d ) .
> 3. For **global** non-linear artifacts, we can set a proper $\mathbf{\sigma_{\mathbf{y}}}$ to cover them.
> 4. For **local** non-linear artifacts, we can directly mask them by hand.
> Hence all we need is to construct $\mathbf{A}=\mathbf{A_{color}}\mathbf{A_{mask}}$ and set a proper $\sigma_{\mathbf{y}}$. We have proved $\mathbf{A_{color}}$ and $\mathbf{A_{mask}}$ and their pseudo-inverse can be easily constructed by hand. We will add a Pytorch-like implementation to Appendix.
>
> We have added some experiment results in Appendix D, where you can see how the real-world noise is removed with different $\sigma_{\mathbf{y}}$.
>
> ###  **Q: "Computing complexity"**
>
> A: Thanks for your advice! Hope the explanation and results in our overall response can solve your concerns.

---

### Author Response · Authors · 2022-11-14
**Overall Response**

## To all reviewers:

We sincerely thank all four reviewers for the constructive feedback! Correspondingly, we have made the following updates to our submission:
1. Add Appendix A (time and memory consumption).
2. Add Appendix B (compare with supervised methods).
3. Add Appendix D (solving real-world degradation using DDNM+).
4. Add Appendix E (PyTorch-like codes).
5. Modify some wording to make the claims more precise.
6. We are working on the corresponding GitHub project. The source code will be made public.

Here we make overall responses to common concerns and then reply to each reviewer's questions in detail.

## **Q1: "Computing complexity"**

A: Our method has obvious advantages in computing complexity among recent zero-shot diffusion-based restoration methods [1,2,6]. These methods are all based on basic diffusion models, the differences are how to bring the constraint $\mathbf{y}=\mathbf{Ax}+\mathbf{n}$ into the reverse diffusion process. We conclude our advantages as below:
1. DDNM yields almost the same cost as the original diffusion models on memory and computation.
2. DDNM does not need any optimization toward minimizing $||\mathbf{y}-\mathbf{A}\mathbf{x}\_{0|t}||$ since we directly yield the optimal solution by range-null space decomposition (Section 3.1) and precise range-space denoising (Section 3.3). We notice some recent works [2,6,7] resort to such optimization, e.g., $\mathbf{x}\_{t-1} = \mathbf{x}\_{t-1} - \zeta_t\nabla_{\mathbf{x}\_{t}}||\mathbf{y}-\mathbf{A}\mathbf{x}\_{0|t}||^2_2$. But these methods suffer costly gradient computation. Experiment on 4x SR task shows that DDNM yields superior results with significantly less cost on memory and time.
3. Unlike [1,3], DDNM does not necessarily need SVD[1] or Fourier transform[3]. As is presented in Section 3.2, we construct $\mathbf{A}$ and $\mathbf{A}^{\dagger}$ for colorization, inpainting, and super-resolution problems **by hand**, which bring negligible computation and memory consumption. We also show that these hand-designed operator works well and can be combined to solve complex tasks. In contrast, SVD-based methods suffer heavy costs on memory and computation if $\mathbf{A}$ has a high dimension (e.g., 128xSR, as shown below).

Below experiments well support these claims:

\*experiment setting：For all methods, we use the same pretrained model from [256x256 diffusion (not class conditional)](https://openaipublic.blob.core.windows.net/diffusion/jul-2021/256x256_diffusion_uncond.pt) on a single RTX 2080Ti GPU. Test images are from this [ImageNet 1K testset list](https://github.com/XingangPan/deep-generative-prior/tree/master/scripts). We use the average-pooling downsampler, 4xSR, 100 DDIM[5] steps with η=0.85 and without classifier guidance. For DPS\[6], we set $\zeta_t=100\sqrt{\bar{\alpha}\_{t-1}}$.

| Method |  inference time (s/image) | Memory (MB) | PSNR↑/FID↓ |
|  :-  |  :-  | :-  | :-  |
|  DDPM[4] (unconditional)  | 11.9 | 5758 | N/A |
|  DPS[6]  | 36.5 | 8112 | 25.51/55.92 |
|  DDRM[1]  | 12.4 | 5788 | **27.05**/38.05 |
|  **DDNM** (ours)  | **11.9** | **5728** | 27.04/**33.81** |

\*experiment setting：The same as above, except for the SR scale.
| Method |  inference time (s/image) | Memory (MB) |
|  :-  |  :-  | :-  |
|  DDRM\[1](64xSR)  | 36.4 | 5788 |
|  **DDNM**(64xSR)  | **11.9** | **5728** |
|  DDRM\[1](128xSR)  | 83.3 | 6792 |
|  **DDNM**(128xSR)  | **11.9** | **5728** |


## **Q2: "Comparing to supervised methods"**

A: Our method is superior to existing supervised IR methods [8,9] in these ways:
1. DDNM is zero-shot for diverse tasks, but supervised methods need to train separate models for each task.
2. DDNM is robust to degradation modes, but supervised methods own poor generalized performance.
3. DDNM yields significantly better performance on certain datasets and resolutions (e.g., ImageNet at 256x256).

These claims are well supported by below experiments:

\*experiment setting： DDNM uses the same setting as Q1 unless otherwise specified. [SwinIR-L](https://github.com/JingyunLiang/SwinIR/releases/download/v0.0/003_realSR_BSRGAN_DFOWMFC_s64w8_SwinIR-L_x4_GAN.pth)\[8] and [BSRGAN](https://github.com/cszn/BSRGAN)\[9] are official pretrained models.

| Method | (bicubic, $\sigma_{\mathbf{y}}$=0 ) PSNR↑/FID↓ | (average-pooling, $\sigma_{\mathbf{y}}$=0 ) PSNR↑/FID↓ | (average-pooling, $\sigma_{\mathbf{y}}$=0.2 ) PSNR↑/FID↓ |inference time (s/image)|
|  :-  |  :-  | :-  | :-  | :-  |
|  SwinIR-L[8] |  21.21/56.77 | 23.88/54.93 | 18.39/134.18 | 6.1 |
|  BSRGAN[9]  | 21.46/68.15 | 24.14/67.70 | 14.06/195.41 | **0.036** |
|  **DDNM**(ours)  | **27.46/39.26** | **27.04/33.81** | **22.67/80.69** | 11.9 |

We do admit that DDNM has some drawbacks compared to supervised IR methods. However, these drawbacks are usually originated from the deficiency of diffusion models:
1. Diffusion models have slow inference speed.
2. Diffusion models struggle in synthesizing high-resolution images, e.g., FFHQ at resolution 1024x1024.

---

> ### Author Response · Authors · 2022-11-14
> **References**
>
> **References:**
>
> >[1] Kawar et al., Denoising diffusion restoration models, NeurIPS 2022.
> >
> >[2] Ho et al., Video diffusion models, arXiv preprint arXiv:2204.03458
> >
> >[3] Song et al., Solving inverse problems in medical imaging with score-based generative models, ICLR 2022.
> >
> >[4] Ho et al., Denoising diffusion probabilistic models, NeurIPS 2020.
> >
> >[5] Song et al., Denoising diffusion implicit models, ICLR 2021.
> >
> >[6] Chung et al., Diffusion posterior sampling for general noisy inverse problems, arXiv preprint arXiv:2209.14687.
> >
> >[7] Chung et al., Improving diffusion models for inverse problems using manifold constraints, NeurIPS 2022.
> >
> >[8]Liang et al., Swinir: image restoration using swin transformer, ICCVW 2021.
> >
> >[9]Zhang et al., Designing a practical degradation model for deep blind image super-resolution, ICCV 2021.
> >
> >[10]Choi et al., ILVR: conditioning method for denoising diffusion probabilistic Models, ICCV 2021.
> >
> >[11]Lugmayr et al., Repaint: Inpainting using denoising diffusion probabilistic models, CVPR 2022.

---

### Public Comment · ~Zhengxiong_Luo1 · 2023-02-03
**Eq (13) may destroy the formulation of diffusion models**

In original diffusion models, Eq (12) is immediately followed by Eq (14). While in this paper, Eq (13) is inserted between them, which may destroy the formulation of diffusion models. Actually, I think Eq (14) does not hold there.

Also, according to the discussions in Sec 3.1, the realness will be strictly kept in the proposed DDNM. However, Fig 3 shows that it is not the case. In fact, the realness will also be lost in Eq (14). Although \hat{x}{0|t} is subject to the constraint of realness, x{t-1} is not. If the realness cannot be kept, then what is the superiority of this DDNM against ordinary SR diffusion models?

I wish someone could help me figure out these questions.

---

> ### Author Response · Authors · 2023-02-03
> **Reply to Zhengxiong Luo**
>
> That's a good observation, but I don't think there's a problem with Eq (14) holds or not, since Eq (14) is just a step of adding noise. The core difficulty is in Eq (12): estimating the clean image $x\_{0|t}$ from the previous noisy step $x\_{t}$ using a pre-trained denoiser. When $x\_{t}$ is well within the distribution $q(x\_{t})$ that the denoiser is trained on, Eq (12) works well in practice (e.g., in low-scale SR). But when $x\_{t}$ is obviously out of the distribution $q(x\_{t})$, DDNM yields unreal results, and that's why we use the time-travel trick to improve realness (e.g., in low sampling ratio CS).
>
> You may reference the section "Time-Travel For Better Restoration Quality" for details. Or, get more intuitive understanding by observing $x\_{t}$, $x\_{0|t}$, and $\hat{x}\_{0|t}$ in the video on our project page: https://wyhuai.github.io/ddnm.io/.

---

### Decision · Program_Chairs · 2023-01-20

**Decision:**

Accept: notable-top-25%

**Justification For Why Not Higher Score:**

This paper is a novel and solid application work for image restoration tasks. Considering that it is an application paper, and the potential impact might be limited, that is to say, the paper doesn't provide a fundamental contribution or make a breakthrough in the field, thus, the AC recommends "accept with spotlight".

**Justification For Why Not Lower Score:**

All 4 reviewers recommend accepting the paper, and 3 of them assign a score 8: "Accept".

**Metareview: Summary, Strengths And Weaknesses:**

This paper presents a novel zero-shot framework for linear image restoration, and the key component is a denoising diffusion null-space model. The framework only needs a pre-trained diffusion model, and doesn’t require further training and optimization. Experimental results show that the proposed method outperforms state-of-the-art zero-shot restoration methods in many tasks. The proposed framework in the paper is novel and interesting. Besides, the experiments are comprehensive and solid.  The review process is summarized as below: The paper eventually received a total of 4 positive reviews: 3 “Accept” and 1 “marginally above the acceptance” after the rebuttal. The rebuttal successfully addresses all the reviewers’ concerns, and all reviewers agree on accepting the paper. Authors provided additional experimental results to further strengthen the paper, and the revised version has been improved after taking into account all the comments and suggestions from reviewers. After reading the rebuttal and reviewers’ comments, the AC agrees with the reviewers and recommends accepting the paper because of its novelty, value, and solid experiments.

**Note From Pc:**

if the above contains the word "oral" or "spotlight" please see: "oral" presentation means -> notable-top-5% and "spotlight" means -> notable-top-25%. As stated in our emails, we are disassociating presentation type from AC recommendations